# Coordination of virulence factors and lifestyle transition in *Pseudomonas aeruginosa* through single-cell analysis
Haozhe Chen[1], Gabriele Malengo [2], Liyun Wang[3], Olga Vogler[4], Mandy Renner[4,5], Timo Glatter[6], Nicole Paczia[7], Daniel Unterweger[4,5], Victor Sourjik[3] & Andreas Diepold [1,8]✉

*Pseudomonas aeruginosa*, a versatile Gram-negative opportunistic pathogen, relies on multiple virulence mechanisms, including a Type III Secretion System (T3SS) and several Type VI Secretion Systems (T6SS), to establish infections. The bacterial universal second messenger cyclic di-guanylate (c-di-GMP) orchestrates the lifestyle transitions of *Pseudomonas aeruginosa* between motile and biofilm-associated states and influences the expression of virulence traits. While it is clear that these systems are interconnected, their precise interaction on the single-cell level has remained unclear. In this study, we use single-cell analysis to dissect the role of c-di-GMP in the heterogeneity of virulence factors in *P. aeruginosa* populations. Our results confirm earlier findings that on the population level, high c-di-GMP levels lead to increased formation and activity of the H1-T6SS, while negatively influencing formation and activity of the T3SS. On the single-cell level, we further characterize the virulence crosstalk within *P. aeruginosa* populations by presenting a cooperative relationship among T3SS and flagellum and antagonistic relationships between presence of the H1-T6SS and the T3SS as well as the flagellum. Overall, this c-di-GMP-orchestrated heterogeneity and crosstalk of virulence systems suggest a strategy to optimize survival and pathogenicity under varying environmental conditions in the framework of the motile-sessile lifestyle transition.

*Pseudomonas aeruginosa* is a Gram-negative, rod-shaped bacterium and an opportunistic pathogen linked to numerous infections in immunocompromised individuals, such as those with cystic fibrosis, cancer, or burn wounds[1–3]. Clinical infections caused by *P. aeruginosa* are often chronic and associated with high mortality rates, primarily due to the high bacterial resistance to various antibiotics, which complicates targeted treatment efforts[4]. Carbapenem-resistant *P. aeruginosa* is classified as "high priority" on the World Health Organization (WHO) list of bacterial pathogens[5,6], highlighting the global threat it poses and the urgent need to understand its virulence mechanisms.

To survive in complex environments and counter host immune defenses, *P. aeruginosa* has developed a robust arsenal of virulence systems. The Type III Secretion System (T3SS) allows bacteria to deliver specialized effector proteins into the host cell through a needle connecting the bacterial

and host cytosol. Extensive evidence supports the role of the T3SS in breaching the epithelial layer, an essential barrier to pathogens[7,8]. Structurally, the T3SS resembles a syringe-like injection apparatus and is thus also named injectisome. It is conserved across important Gram-negative pathogens[9]. To efficiently infect the host, T3SS secretion is under tight regulation. The transcription of T3SS-related operons is primarily controlled by the master activator ExsA and an ExsC/E/D/B partner-switch complex[10,11]. In the absence of contact with a host cell, the T3SS is maintained in an inactive state[12,13]. Contact with host cells triggers a switch controlled by the "gatekeeper complex," allowing the release of effectors[14–17], which can be mimicked by $Ca^{2+}$ depletion. At the same time, the repressor ExsE is secreted through the T3SS, which releases ExsA and leads to an upregulation of T3SS machinery and effector expression[13]. The main effectors of the *P. aeruginosa* T3SS, ExoS, ExoT, ExoY, and ExoU, disrupt

[1]Max Planck Institute for Terrestrial Microbiology, Department of Ecophysiology, Marburg, Germany. [2]Max Planck Institute for Terrestrial Microbiology, Core Facility for Flow Cytometry and Imaging, Marburg, Germany. [3]Max Planck Institute for Terrestrial Microbiology, Center for Synthetic Microbiology (SYNMIKRO), Marburg, Germany. [4]Max Planck Institute for Evolutionary Biology, Plön, Germany. [5]Institute for Experimental Medicine, Kiel University, Kiel, Germany. [6]Max Planck Institute for Terrestrial Microbiology, Core Facility for Mass Spectrometry and Proteomics, Marburg, Germany. [7]Max Planck Institute for Terrestrial Microbiology, Core Facility for Metabolomics and Small Molecule Mass Spectrometry, Marburg, Germany. [8]Department of Applied Biology, Institute of Applied Biosciences, Karlsruhe Institute of Technology (KIT), Karlsruhe, Germany. ✉e-mail: andreas.diepold@kit.edu

host cell membranes and the actin cytoskeleton through various enzymatic activities, often leading to host cell death[18].

Another critical weapon for sustaining infection is the Type VI Secretion System (T6SS). *P. aeruginosa* possesses four distinct T6SSs, labeled H1-T6SS through H4-T6SS [19,20]. Of these, H1-T6SS is the most studied. It primarily targets prokaryotic cells and mediates inter-bacterial competition[21]. The H1-T6SS has been found to be associated with the biofilm lifestyle by eliminating competing bacteria in mixed species biofilms, which is the most common type in clinical infections[22]. Therefore, this study focuses on the H1-T6SS of *P. aeruginosa*. The H1-T6SS in *P. aeruginosa* operates on a "tit-for-tat" strategy, where it can be activated as a counter-attack measure in response to membrane breaches by competing bacteria[23]. Similar to the T3SS, the T6SS also forms an extended structure, but with a contractile sheath located in the cytosol. When the sheath contracts, an inner tube is expelled and delivers associated effectors into the neighboring cell[24]. To date, eight effectors, Tse1 to Tse8, have been identified in H1-T6SS of *P. aeruginosa*, each paired with an immunity protein (Tsi1 to Tsi8) to prevent self-targeting [19,25,26]. These effectors help *P. aeruginosa* establish survival advantages within complex bacterial communities by degrading the cell walls of competing bacteria or disrupting their signaling pathways to induce bacteriostasis [27–30].

A typical feature of *P. aeruginosa* is its motile-sessile lifestyle transition. This transition is primarily regulated by the bacterial second messenger cyclic diguanylate (c-di-GMP)[31], which regulates biofilm formation[32] and virulence across many bacterial species, including *P. aeruginosa*[33]. c-di-GMP levels are negatively correlated with motility across various organisms[34]. The internal c-di-GMP level is modulated by c-di-GMP metabolic enzymes: diguanylate cyclases (DGCs, for synthesis) and phosphodiesterases (PDEs, for degradation)[35–37]. When mobile *P. aeruginosa* cells detect a landing surface, the c-di-GMP synthase WspR is activated. The elevated c-di-GMP levels halt flagellum biogenesis through the transcriptional regulator FleQ [38–40]. As the cells progress toward irreversible attachment, c-di-GMP promotes biofilm matrix production through its receptor PelD and the regulator FleQ, providing environmental protection for the bacteria [39,41,42]. In this phase, another biofilm matrix component, the exopolysaccharide Psl, can function as an extracellular signal to stimulate c-di-GMP production and reinforce biofilm formation[43]. Consistent with this survival strategy, high levels of c-di-GMP have also been associated with increased antibiotic resistance[44]. In later stages, such as in response to deteriorating environmental conditions, *P. aeruginosa* cells disperse from the biofilm with rejuvenated motility, rejoining the transition cycle. Several phosphodiesterases (PDEs) that decrease c-di-GMP levels play important roles in the biofilm dispersal process[45,46]. This c-di-GMP-mediated cycle switch between motility and biofilm formation enables *P. aeruginosa* to dynamically adapt to changing environments by promoting dispersal under favorable conditions and persistence under stress, thereby promoting its survival.

Does a specific lifestyle correspond to unique virulence traits? Various studies have suggested that acute infections are primarily caused by planktonic cells, while biofilm-associated cells tend to cause persistent infections [47–49]. A key regulator of this switch is the Gac/Rsm signaling pathway. In this pathway, RsmA acts as a transcriptional regulator, promoting traits linked to acute infections, including flagella, type IV pili, and the T3SS, while repressing traits associated with persistent infections, such as biofilm matrix production and T6SS activity [50,51]. Deletion of RsmA has been shown to reduce cytotoxicity in epithelial cells and increase persistence in mouse models of infection[52]. Upstream, sensor kinases like GacS, RetS and LadS participate in Gac/Rsm-mediated modulation, linking environmental cues to physiological changes [50,53–56]. C-di-GMP also contributes to this switch. Moscoso and colleagues found that deletion of *retS* in *P. aeruginosa* leads to increased c-di-GMP levels, which repress the T3SS and upregulate the T6SS. This finding has led to a virulence transition model where c-di-GMP functions as a main agent in the Gac/Rsm-mediated T3SS/T6SS switch[57]. However, recent findings have challenged this model. High c-di-GMP levels in *P. aeruginosa* were found to be associated with decreased H1-T6SS expression and killing efficiency, mediated by the c-di-GMP

receptor FleQ[58]. A similar result has been reported for *P. putida*, another opportunistic pathogen within the *Pseudomonas* family[59]. Given the complexity of *P. aeruginosa* virulence and c-di-GMP signaling, further studies are necessary to establish which signaling pathways mediate the conversion between T3SS-mediated acute infection and biofilm-associated chronic infection in *P. aeruginosa*.

It is commonly believed that bacterial populations exhibit phenotypic heterogeneity for fitness benefits. A paradigm of this is defined in biofilm communities, where a diversity of metabolic capabilities allows the utilization of different nutrient sources. Such heterogeneity is also found in virulence systems[60]. Rather than switching virulence systems on or off across the entire population, *P. aeruginosa* often exhibits a bistable distribution pattern, in which cells are divided into distinct subgroups. The cAMP-Vfr pathway, a global transcriptional regulator, is associated with bistable T3SS expression in *P. aeruginosa*[61], while c-di-GMP promotes flagellum heterogeneity within the community [62,63]. Phenotypic heterogeneity in the bacterial population is generally believed to be associated with either a "division of labor" strategy, where different subpopulations with separated tasks operate cooperatively to bolster the fitness of the entire population with low energy cost; or a "bet hedging" strategy, where the diversification of phenotypes ensures the survival of certain individuals under the selective pressure of fluctuating environments. Under most circumstances, phenotypic heterogeneity is not strictly classified as one strategy or the other, but rather a combination as a joint effort to promote survival[64].

The issue of bistability challenges traditional approaches that treat bacterial populations as homogeneous. It also presents challenges for established c-di-GMP measurement methods, such as metabolomics. C-di-GMP levels have been shown to vary, at times strongly, between neighboring bacteria[65], and phenotypic heterogeneity can have a direct influence on population characteristics[66]. Despite this, the direct link between c-di-GMP levels and virulence factors has not been investigated on the single-cell level. In this study, we used a recently developed Förster resonance energy transfer (FRET) biosensor[67], which allows for the analysis of the c-di-GMP levels in individual bacteria. To better clarify the role of c-di-GMP in *P. aeruginosa* virulence regulation and its influence on the distribution and potential crosstalk among different virulence systems, we conducted functional analyses at the single-cell level. Our findings reveal that c-di-GMP correlates with specific virulence trait distributions in subpopulations. Single-cell microscopy further revealed correlations between the presence of the T3SS, H1-T6SS, and the flagellum. Overall, our results suggest a c-di-GMP-modulated and actively upheld heterogeneity in virulence systems of *P. aeruginosa* on the single-cell level and provide novel evidence for the virulence transition model. These insights deepen our understanding of *P. aeruginosa* as a pathogen and may contribute to developing therapeutic strategies or biomedical applications to manage its infections.

## Results

### C-di-GMP downregulates the T3SS and upregulates the H1-T6SS
Previous evidence on the regulatory role of c-di-GMP in the secretion systems of *P. aeruginosa* focused primarily on the expression level of the respective systems, such as the data acquired from transcriptomic analyses [57,68]. We therefore designed experiments to examine the regulation from a functional perspective, including the c-di-GMP influence on the assembly and activity of secretion systems, in live bacteria. This analysis takes the study closer to conditions relevant to bacterial infections. To manipulate the intrabacterial c-di-GMP levels in a way unlikely to interact with other regulatory networks, we employed heterologous enzymes influencing the c-di-GMP level, the *Escherichia coli* phosphodiesterase YhjH (also named PdeH) and the *Caulobacter crescentus* diguanylate cyclase DgcA, which we expressed from a plasmid in *P. aeruginosa* PAO1 [69,70]. The effectiveness of these manipulations was confirmed by comparative metabolome analysis. Semi-quantitative measurements of intracellular c-di-GMP indicated that, compared to the wild-type control, strains expressing YhjH showed a significant reduction in c-di-GMP levels, while DgcA expression led to a substantial increase (Suppl. Fig. 1).

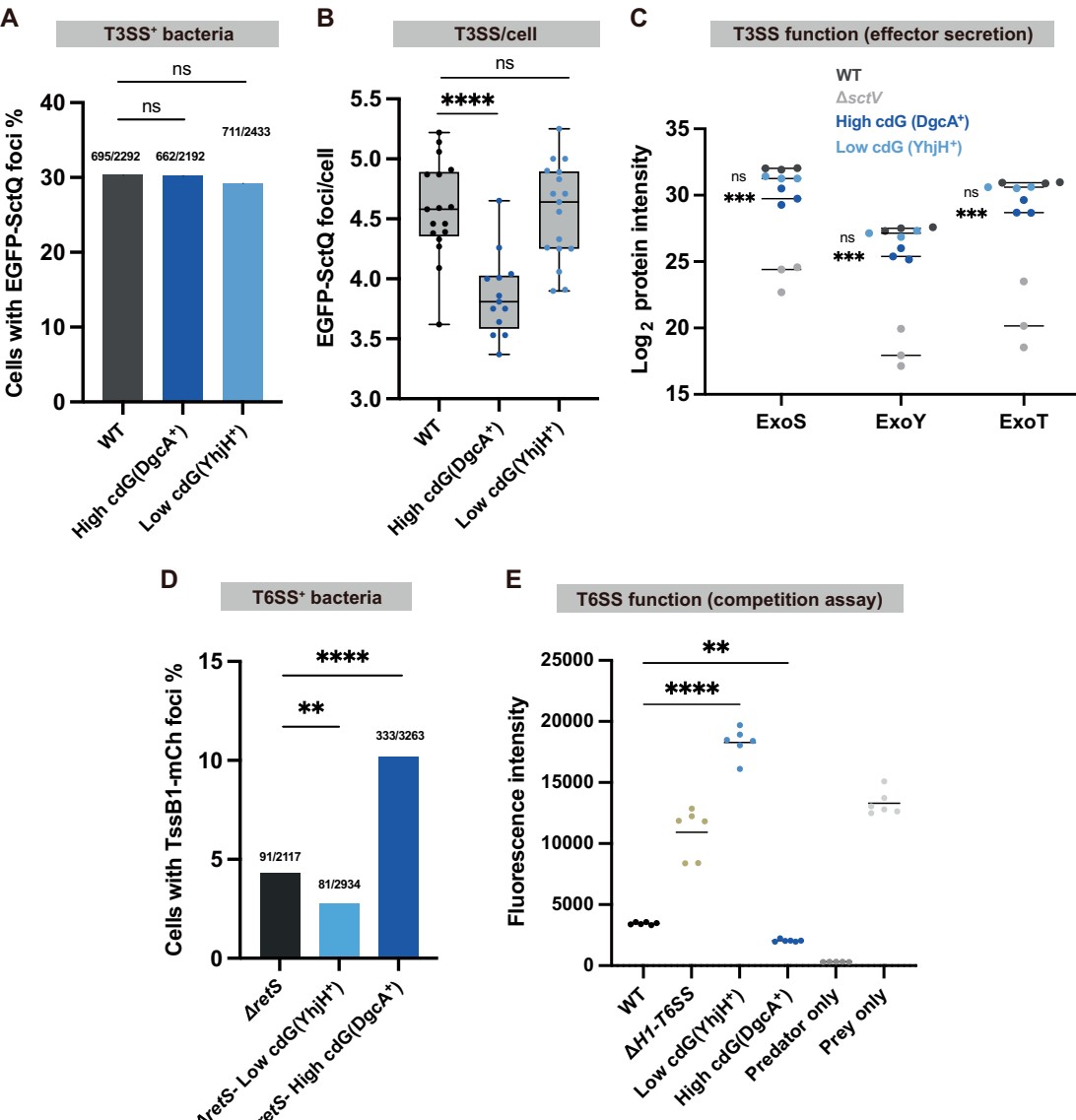

**Fig. 1 | Increased cyclic-di-GMP levels downregulate T3SS assembly and activity, and upregulate H1-T6SS assembly and activity. A** Fraction of T3SS-positive bacteria in different c-di-GMP backgrounds. Data is shown as a fraction of cells with EGFP-SctQ foci; exact numbers are shown above each bar. Samples were taken from >10 different fields of view in 3 different experiments. Statistical analysis was done via $\chi^2$ test, ns, non-significant (the precise numbers of all *p*-values are provided in Suppl. Data 1). **B** Number of T3SS per bacterium in different c-di-GMP backgrounds. Data is shown as the number of EGFP-SctQ foci per T3SS-positive cell, with each data point representing the calculation from one field of view. Samples were taken from >10 different fields within 3 different experiments. Statistical analysis was done via ANOVA with multiple comparisons to WT, ****, $p < 0.0001$; ns, non-significant. **C** T3SS effector secretion in different c-di-GMP backgrounds examined by label-free quantitative mass spectrometry. ExoS/Y/T are the three T3SS effectors in PAO1; protein intensities (reflecting the abundance of a specific protein in a sample) in the supernatant are compared in different c-di-GMP backgrounds. Data

is shown as log$_2$ mean intensity of effector proteins for the different background strains. Bars indicate the mean value. Biological replicates, $n = 3$. Statistical analysis was done via ANOVA with multiple comparisons to WT, ***, $p < 0.001$; ns, non-significant. **D** Fraction of H1-T6SS-positive bacteria in different c-di-GMP backgrounds. Data is shown as a fraction of cells with H1-T6SS assembly (TssB1-mCherry foci); exact numbers are shown above each bar. Samples were taken from >10 different fields of view in 3 different experiments. Statistical analysis was done via $\chi^2$ test, **, $p < 0.01$; ****, $p < 0.0001$. **E** H1-T6SS killing efficiency test in different c-di-GMP backgrounds. Indicated strains were used as predators, while a YFP-labeled H1-T6SS effector-immunity pair deletion strain ($\Delta tse6tsi6$) was used as prey. Data is shown as background-corrected prey fluorescence. Data points are shown as an endpoint value of a 24-h killing assay. Biological replicates, $n = 3$. Statistical analysis was done via ANOVA with multiple comparisons to WT, **, $p < 0.01$; ****, $p < 0.0001$.

We first tested the influence of varying c-di-GMP levels on the assembly of both secretion systems. For the T3SS, the dynamic cytosolic T3SS component SctQ (PscQ in *P. aeruginosa*-specific nomenclature, see Suppl. Table 1), which plays a role in effector recognition and selection [71,72], was N-terminally labeled with EGFP by allelic exchange, leaving the expression under the native promoter at the original *sctQ* site (Suppl. Fig. 2A). While only a few bacteria were T3SS-positive in non-secreting conditions (high [Ca$^{2+}$]), around 30% of bacteria were T3SS-positive under

secreting conditions (low [Ca$^{2+}$]). This fraction was constant across strains with different c-di-GMP levels, suggesting that c-di-GMP does not influence the overall T3SS bistability (Fig. 1A). However, increased c-di-GMP levels led to a lower number of injectisomes per cell, with 3.86 (95% CI [3.66, 4.07]) foci per cell in the high c-di-GMP strain, compared to 4.58 (95% CI [4.37, 4.79]) in the plasmid control and 4.56 (95% CI [4.35, 4.77]) foci per cell in the lower c-di-GMP strain, showing a small, but meaningful repressive effect of increased c-di-GMP on T3SS assembly (effect size of Hedge's g of -1.87 for

increased c-di-GMP and of -0.06 for decreased c-di-GMP) (Fig. 1B and Suppl. Fig. 3). To directly test the effect of c-di-GMP on the function of the T3SS, effector secretion, we then analyzed the culture supernatant from each c-di-GMP level strain by proteomics to detect T3SS-specific effectors. The results show that high c-di-GMP levels strongly suppress secretion, with a four-fold decrease in the levels of all three tested T3SS effectors in the high c-di-GMP strain (Fig. 1C), corroborating the assembly data. They are also in line with previous findings that the low level of c-di-GMP in bacteria dispersed from biofilms helps to increase T3SS-mediated cytotoxicity[73]. Collectively, our analysis demonstrates that c-di-GMP downregulates T3SS function, in line with previous data[57].

Subsequently, the assembly of H1-T6SS at different c-di-GMP levels was measured using a similar workflow. The H1-T6SS assembles on demand, and its assembly-disassembly cycle is completed within minutes, which makes H1-T6SS assembly also a readout for the activity of the system[74]. The H1-T6SS sheath protein TssB1 was C-terminally tagged with mCherry by allelic exchange (Suppl. Fig. 2B). Preliminary experiments showed that the H1-T6SS was not assembled in the absence of competitors. This may be attributed to the "Tit for Tat" behavior of H1-T6SS in *P. aeruginosa*, which responds to membrane breaches[23]. To be able to investigate T6SS assembly in the absence of competing bacteria, RetS—a sensor kinase known to repress H1-T6SS expression and decrease H1-T6SS assembly rate[75,76]—was deleted. This modification increased the H1-T6SS-positive cell population and allowed to detect a clear positive correlation between c-di-GMP levels and H1-T6SS assembly (Fig. 1D). A competition assay confirmed the supportive role of c-di-GMP in H1-T6SS function. A YFP-labeled H1-T6SS effector-immunity pair deletion strain (Δtse6tsi6) served as the prey, while strains with different c-di-GMP backgrounds acted as predators. Tracking prey fluorescence over 24 h showed that predators with low c-di-GMP levels had a significantly reduced killing efficiency (indicated by a higher YFP fluorescence as readout for the prey survival rate), whereas high c-di-GMP levels reduced the survival of prey cells to less than half compared to WT (Fig. 1E and Suppl. Fig. 4).

## Presence of the T6SS, but not the T3SS correlates with increased cellular c-di-GMP level

Previous studies indicated that c-di-GMP targets can, in turn, regulate c-di-GMP levels, leading to positive feedback loops. In *P. aeruginosa*, the flagellar stator MotC of *P. aeruginosa* has been shown to stimulate c-di-GMP production by binding and activating the diguanylate cyclase (DGC) SadC[77]. To investigate whether secretion systems can also affect c-di-GMP levels, we deleted SctV (PcrD in *P. aeruginosa*-specific nomenclature), an essential component of the T3SS export apparatus, without which no complete injectisomes can be formed. Additionally, we generated a ΔH1-T6SS strain with a deletion of the complete coding region for H1-T6SS, which spans from PA0071 to PA0091.

To test c-di-GMP levels in the T3SS/H1-T6SS-deficient strains, we employed a recently developed Förster resonance energy transfer (FRET) c-di-GMP reporter system[67], which allows for real-time, non-destructive quantification of c-di-GMP at single-cell resolution without the need for an external normalization fluorophore. The plasmid-based reporter was designed with the c-di-GMP receptor YcgR sandwiched between a fluorescent donor (mTurquoise2) and an acceptor (mNeonGreen) (Suppl. Fig. 2C). Upon binding of c-di-GMP, YcgR undergoes a conformational change that brings donor and acceptor closer together, enabling fluorescent energy transfer[67]. C-di-GMP levels were quantified by FRET efficiency, measured as the increase in donor signal upon acceptor photobleaching (Suppl. Fig. 5). A control experiment with high and low c-di-GMP strains validated the suitability of the method for use in *P. aeruginosa* (Suppl. Fig. 1B).

We found that the absence of the T3SS had no impact on c-di-GMP levels (Fig. 2A). However, a significant decrease in c-di-GMP levels (Cohens' $d = -0.74$) was observed in the ΔH1-T6SS strain, indicating that the presence of the H1-T6SS increases intracellular c-di-GMP levels (Fig. 2B). This is in line with the decreased biofilm formation in a T6SS-deficient strain (Suppl. Fig. 6).

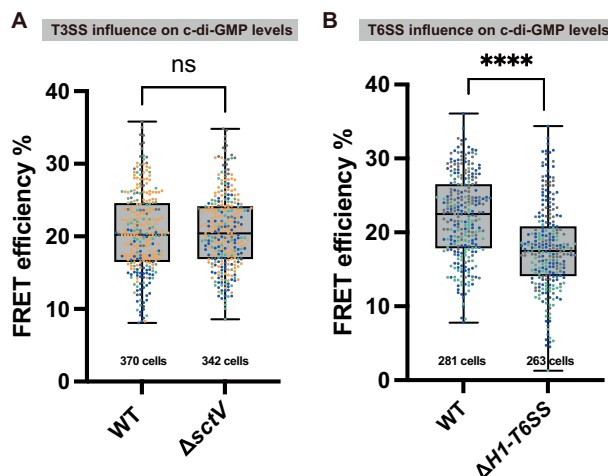

**Fig. 2 | Presence of the H1-T6SS, but not T3SS, leads to increased c-di-GMP levels. A** FRET measurement of c-di-GMP between a T3SS-deficient strain (ΔsctV) and WT. Data is shown as corrected FRET efficiency, a positively correlated indicator of c-di-GMP levels, see Suppl. Fig. 1. Detailed calculations see Methods. Each data point represents a measurement of a single cell. Experiments were repeated on multiple dates, with each color set representing data points from one day. Highest and lowest end of each bar represent maximum and minimum value, respectively; boxes indicate median and 25th–75th percentile. The total cells tested for each group is indicated under the respective bars. **B** FRET measurement of c-di-GMP between a H1-T6SS deletion strain (ΔH1-T6SS) and WT. Graph explanation is the same as above. Statistical analysis was done via Student's *t*-test; ns, non-significant; ****, $p < 0.0001$.

## Stochastic heterogeneity of cellular c-di-GMP levels influences secretion systems subpopulations

C-di-GMP plays a crucial role in coordinating the switch between infection modes, as evidenced by its regulatory effects on T3SS and H1-T6SS gene expression (Fig. 1)[57]. Given the bistability of the secretion systems and the heterogeneous distribution of c-di-GMP levels among different cells[78], we employed single-cell FRET microscopy to better understand the involvement of c-di-GMP in this process.

T3SS secretion is accompanied by strongly increased expression of T3SS effectors. To visualize T3SS activity at the single-cell level, we therefore designed a plasmid-based T3SS functional reporter that expresses mCherry under the control of the promoter for *exoS*, the most strongly expressed T3SS effector protein in *P. aeruginosa*[18]. The fluorescence intensity of the resulting P_{exoS}-mCherry strain serves as a readout for T3SS activity (Suppl. Fig. 7), and we found that high reporter activity coincided with T3SS assembly on the single-cell level (Suppl. Fig. 8). Analysis of *exoS* expression revealed a negative correlation between c-di-GMP levels and T3SS activity: c-di-GMP levels were higher in the T3SS-inactive subpopulation, whereas the T3SS-active subpopulation had lower c-di-GMP levels (Cohens' $d = -0.57$) (Fig. 3A).

To distinguish H1-T6SS-positive and -negative subpopulations, we analyzed the assembly of the T6SS sheath with the TssB1-mCherry fusion in a ΔretS background, and classified cells with H1-T6SS foci as T6SS-positive same as in Fig. 1D. Opposite to what was observed for the relation between c-di-GMP and the T3SS, c-di-GMP levels were higher in the T6SS-positive subpopulation than in the T6SS-negative one, indicating a positive correlation between c-di-GMP and T6SS activity (Cohens' $d = 0.60$) (Fig. 3B). Considering that c-di-GMP acts as a switch for bacterial lifestyles, this opposite correlation suggests that the two secretion systems are primarily activated in different environments or populations.

## T3SS and H1-T6SS are antagonistic at the single-cell level

As shown above, the T3SS and H1-T6SS exhibit bistable distribution patterns, which were influenced oppositely by c-di-GMP levels at the single-cell level. This negative correlation indicates possible crosstalk between the

**Fig. 3 | C-di-GMP levels are negatively correlated with T3SS activity and positively correlated with H1-T6SS activity on the single-cell level. A** FRET measurement of c-di-GMP in T3SS-negative and -positive populations (T3SS-, T3SS+). Data is shown as corrected FRET efficiency as in Fig. 2, detailed calculations, see "Methods". The microscopy image below with merged fluorescence channels illustrates the classification of T3SS-positive cells by fluorescence of the $P_{exoS}$-mCherry reporter for T3SS activity. **B** FRET measurement of c-di-GMP in T6SS-negative and -positive populations (T6SS-, T6SS + ). The microscopy image below indicates the classification of T6SS-positive cells by the presence of TssB1-mCherry structures. Cell outlines are indicated by white lines. In both graphs, each data point represents a measurement of a single cell. Experiments were repeated on multiple dates, with each color set representing data points from one day. Highest and lowest end of each bar represent maximum and minimum value, respectively; boxes indicate median and 25th–75th percentile. The number of cells tested for each group is indicated below. See Suppl. Fig. 9 for larger fields of view. Statistical analysis by Student's $t$-test, ***, $p < 0.001$; ****, $p < 0.0001$.

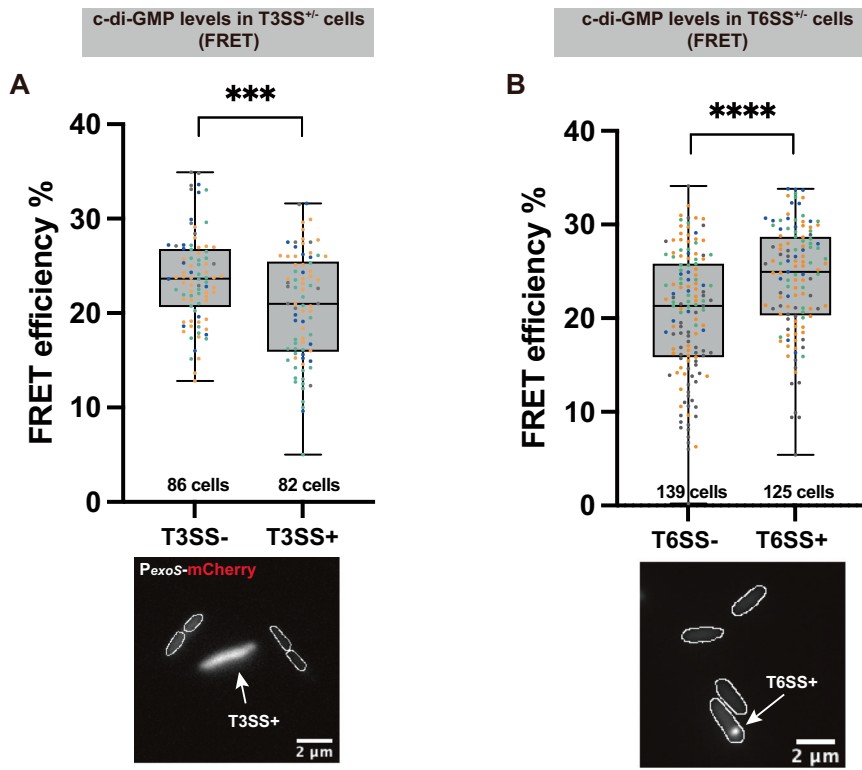

systems. To test this hypothesis, we investigated the correlation between the T3SS and the T6SS in single-cell microscopy. A noteworthy drop in H1-T6SS assembly was observed when culturing the bacteria in T3SS-secreting conditions (low $[Ca^{2+}]$) (Fig. 4A and Suppl. Fig. 10). This was corroborated by a proteomics analysis. Samples cultured under T3SS-secreting conditions showed an eight-fold decrease in H1-T6SS secreted protein levels compared to those cultured in T3SS-non-secreting conditions (high $[Ca^{2+}]$) (Suppl. Fig. 11).

To study the relation between the T3SS and the T6SS, we used a strain where the T3SS can be activated without influencing culture conditions by deleting the T3SS gatekeeper protein SctW (PopN in *P. aeruginosa*-specific nomenclature), and the T6SS-positive subpopulation is increased by the absence of RetS (Suppl. Fig. 12). Microscopy of the T3SS/H1-T6SS dual-labeled strain (EGFP-SctQ, TssB1-mCherry) in this Δ*sctW* Δ*retS* background revealed a notable separation in the distribution of fluorescent signals representing the respective secretion systems (Fig. 4B). Detailed analyses showed that the overall TssB1-mCherry fluorescence intensity was strongly decreased (Cohens' $d = -0.83$) in cells with assembled T3SS compared to those without (Fig. 4B). The reduced expression of this key H1-T6SS component in the T3SS-active subpopulation suggests an antagonistic relationship between these two secretion systems. In line with this finding, we detected significantly fewer bacteria that were both T3SS-positive and T6SS-active than expected from the individual proportions of both populations. A contingency test confirmed a direct antagonism between T3SS and H1-T6SS (Fig. 4C). This antagonism provides additional evidence for a strategy of virulence specialization within the population. In combination with the previous findings that T3SS secretion decreased H1-T6SS effector secretion (Suppl. Fig. 11) and that T3SS secretion decreased biofilm formation, this indicates that T3SS activity might act as a signaling event in the motile-sessile transition in *P. aeruginosa*.

### Flagellum assembles cooperatively with T3SS but antagonistically with H1-T6SS

The presence of flagella is a main factor in the motile-sessile lifestyle transition in *P. aeruginosa*, and is itself linked with other virulence traits in *P.*

*aeruginosa* [79,80]. Therefore, we also considered the flagellum, a crucial structure for mobility, in our study. To this aim, the flagellar motor protein FliM, a homolog of SctQ, was C-terminally labeled with mCherry by allelic exchange. The resulting FliM-mCherry fusion is stable (Suppl. Fig. 2D) and does not impede swimming (Suppl. Fig. 13). Our analysis showed that c-di-GMP levels were substantially lower in cells with an assembled flagellum (Cohens' $d = -0.70$) (Fig. 5A). This finding aligns with previous reports indicating that c-di-GMP negatively regulates flagellum biosynthesis and mobility [81], further supporting a negative correlation between c-di-GMP levels and flagellum assembly.

To better link the activity of secretion systems to the motile-sessile transition in *P. aeruginosa*, we further examined the relationship between the flagellum, the T3SS, and the H1-T6SS at the single-cell level. We created a strain expressing fluorescent fusions to both the flagellum (FliM-mCherry) and T3SS (EGFP-SctQ) by allelic exchange, and cultured these bacteria under T3SS-secreting conditions. Single-cell fluorescence microscopy analysis showed a higher double-positive population than statistically expected, indicating a collaborative interaction between the flagellum and T3SS (Fig. 5B), despite a lack of influence of T3SS presence on swimming (Suppl. Fig. 14).

To complete the correlation circle, a flagellum/H1-T6SS dual-labeled strain (FliM-mCherry, TssB1-Ypet, Suppl. Fig. 2E) in a Δ*retS* background was cultured in LB medium and visualized via microscopy. The analysis revealed a negative correlation, with a substantially lower double-positive population than statistically expected (Fig. 5C). This negative correlation between H1-T6SS and the flagellum mirrors the antagonistic relationship observed between H1-T6SS and T3SS. In summary, the correlations among the three investigated virulence systems provide compelling evidence of the coordinated regulation of virulence across different bacterial lifestyles, highlighting an efficient strategy for infection.

### Discussion

As a model pathogen in microbiology studies, *P. aeruginosa* is known for its c-di-GMP-mediated lifestyle transitions and wide range of virulence factors, such as secretion systems. In recent years, significant progress has been

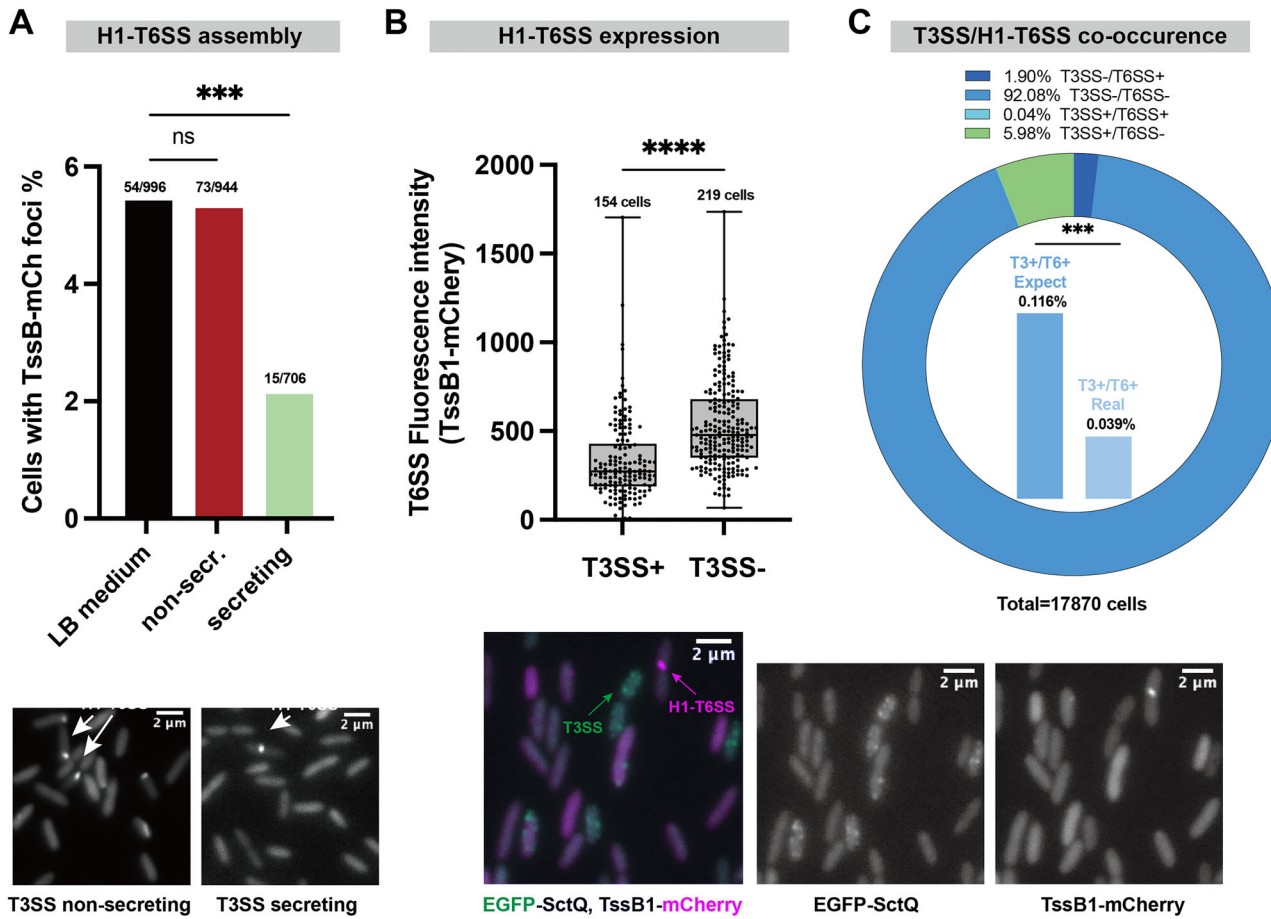

**Fig. 4 | T3SS and H1-T6SS show antagonistic behavior on the single-cell level. A** Microscopy analysis of H1-T6SS assembly in LB, non-secreting (LB + 5 mM CaCl$_2$) and secreting medium (LB + 5 mM EGTA, see Methods for details). Data is shown as a fraction of T6SS-positive bacteria (presence of TssB1-mCherry foci), with exact numbers shown above each bar. Samples were taken from >10 different fields in 3 different experiments. Statistical analysis was done via Student's *t*-test, ***, *p* < 0.001; ns, non-significant. **B** Microscopy analysis T6SS expression levels in EGFP-SctQ TssB1-mCherry Δ*sctW* Δ*retS* bacteria with or without T3SS assembly. Data is shown as background-subtracted and normalized TssB1-mCherry overall fluorescence intensity. Each data point represents the background-corrected TssB1-mCherry overall fluorescence intensity from one cell. Samples were taken from >10 different fields of view from 3 different experiments. Highest and lowest end of each bar represent maximum and minimum value, respectively; boxes indicate median

and 25th–75th percentile. Statistical analysis was done via Student's *t*-test, ****, *p* < 0.0001. Microscopy images below show an overlay of the T3SS channel (green) and the H1-T6SS signal (red), as well as the individual channels (left, green; right, red). **C** Quantification of co-occurrence of T3SS and H1-T6SS in EGFP-SctQ, TssB1-mCherry, Δ*retS*, Δ*sctW* cultured in LB medium. Data is shown as a fraction of bacteria with the indicated presence or absence of the T3SS and T6SS, as evaluated by the presence of EGFP-SctQ and TssB1-mCherry foci; the exact fraction for each group is shown at the top, and the total number of cells tested is indicated at the bottom. Inner histograms show the actual ratio of cells with both T3SS and H1-T6SS, and its comparison to the theoretical ratio based on the individual fractions, assuming no correlation. Samples were taken from >50 different fields in 8 different experiments. Statistical analysis was done via χ² test, ***, *p* < 0.001.

made in understanding the transition between T3SS-mediated acute infection and biofilm-associated chronic infection. Novel regulators, such as the NAD homeostasis modulator NrtR and the RNA ligase RtcB, have been shown to influence virulence traits involved in this transition [82,83]. However, little is known about the role of c-di-GMP in this process, as various studies have led to different conclusions [57–59].

The T3SS was found to be bistably expressed in several bacterial species, with varying fractions of T3SS-positive cells. While *Yersinia enterocolitica* was found to be uniformly T3SS-positive, less than a third of bacteria assembled injectisomes in *P. aeruginosa* [84]. Interestingly, we found that c-di-GMP does not control the overall fraction of T3SS-positive *P. aeruginosa*, but instead reduces the number of T3SS per cell under secreting conditions and the secretion of effectors (Fig. 1). No direct link between c-di-GMP levels and the main regulator of the T3SS in *P. aeruginosa*, ExsA, is known. However, c-di-GMP has been shown to suppress the Vfr-cAMP signaling pathway, which induces *exsA* transcription at the *exsA* promoter [61,85,86]. Increased c-di-GMP levels may therefore reduce the supply of T3SS components and effectors, which could explain the decreased T3SS assembly and secretion observed: While both

expression and assembly of the T3SS and expression and export of effectors are influenced by the same factors, the effect of regulatory cues on effector synthesis is often much stronger[72], in line with the stronger effect on secretion (Fig. 1B, C). Additional c-di-GMP regulation mechanisms, such as a direct influence on the secretion process, might be involved in this process.

Microscopy analysis showed that c-di-GMP positively influences H1-T6SS assembly (Fig. 1D). This result was further validated by an H1-T6SS-mediated killing assay. These findings provide support for the model provided by ref. 57, in which high c-di-GMP represses the T3SS, but upregulates the H1-T6SS. FleQ, the transcriptional regulator responsible for c-di-GMP-mediated H1-T6SS suppression[58] may only respond to specific environmental signals, which remain to be explored. As the two DGCs DgcA, used in this study, and WspR, which was the DGC shown to have an effect on the secretion systems in ref. 57, only share 38% sequence identity, c-di-GMP itself is likely to be the factor influencing *P. aeruginosa* virulence. Although the Gac/Rsm pathway has been identified as necessary for this c-di-GMP regulation[57], the detailed mechanism of this complex regulatory network is still unknown.

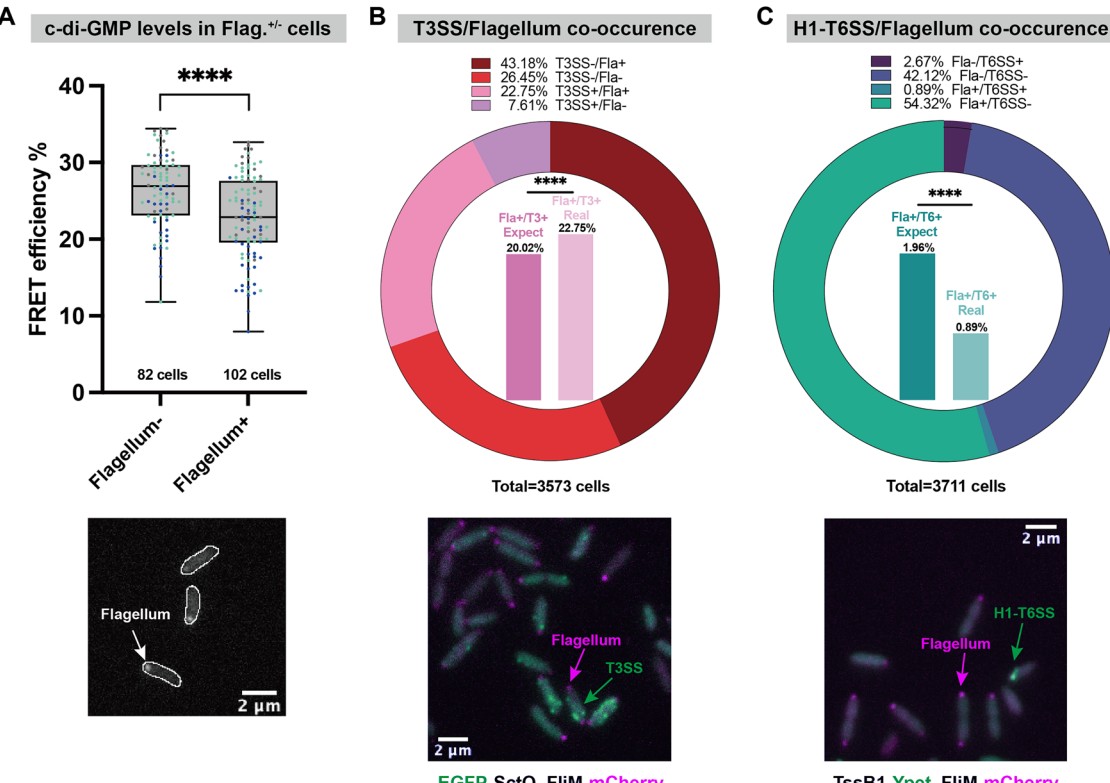

**Fig. 5 | Presence of flagella positively correlates with presence of the T3SS, but negatively with H1-T6SS presence and c-di-GMP distribution. A** FRET measurement of c-di-GMP within flagellum on/off populations. Data is shown as corrected FRET efficiency as in Fig. 2, detailed calculations, see "Methods". Each data point represents a measurement of a single cell. Experiments were repeated on multiple dates, with each color set representing data points from one day. Highest and lowest end of each bar represent maximum and minimum value, respectively; boxes indicate median and 25th–75th percentile. The total cells tested for each group is labeled under the respective bars. Statistical analysis was done via Student's *t*-test, ****, *p* < 0.0001. The microscopy image below indicates flagellum fluorescence label at the cell terminal. Cell outlines are labeled. **B** Quantification of co-existence of T3SS and flagellum (presence of EGFP-SctQ and FliM-mCherry foci, respectively, bacteria cultured in T3SS-secreting medium). Data is shown as a ratio of cells with indicated status (flagellum on/off and T3SS on/off) to total cells the exact ratio for each group is shown at the top, and the total cell amount tested is marked at the bottom. Inner histograms show the actual ratio of cells with both T3SS and flagellum, and its comparison to the theoretical ratio based on the individual fractions. Representative microscopy image below. Samples were taken from >30 different fields and in 3 different experiments. Statistical analysis was done via χ² test, ****, *p* < 0.0001. **C** Quantification of co-existence of H1-T6SS and flagellum (presence of TssB1-Ypet and FliM-mCherry foci, respectively, in a *ΔretS* background, bacteria cultured in LB medium). Graph structure as in (**B**). Representative microscopy image below. Samples were taken from >30 different fields and in 5 different experiments. See Suppl. Fig. 9 for larger fields of view. Statistical analysis was done via χ² test, ****, *p* < 0.0001.

Most studies have investigated the downstream effects of c-di-GMP, given its role as a universal signaling molecule. However, our data shows that H1-T6SS presence also enhances c-di-GMP levels in *P. aeruginosa* (Fig. 2B), similar to earlier findings for the H3-T6SS in *P. aeruginosa*[87]. We propose that this bidirectional regulation functions as a positive feedback loop with a yet unknown molecular basis. In this system, H1-T6SS expression utilizes elevated c-di-GMP levels to reinforce its dominance, establishing bistability. This adaptive strategy also suggests a link between H1-T6SS and the biofilm lifestyle, allowing *P. aeruginosa* to optimize its virulence mechanisms based on environmental needs.

Earlier studies reported a bistable expression pattern of T3SS in *P. aeruginosa*[88], which was also found for the T3SS, as well as the H1-T6SS, in our study. It remains unclear how differences in virulence and bistability are interconnected at the single-cell level. A novel FRET-based c-di-GMP reporting system[67] allowed us to perform a more precise single-cell analysis. Our results show that c-di-GMP levels are positively associated with the presence of the H1-T6SS, while T3SS and flagellum presence are negatively associated with c-di-GMP levels (Figs. 3 and 5A). Notably, both active and inactive states of these virulence systems were observed in cells with intermediate levels of c-di-GMP. Given the complexity of c-di-GMP signaling, it is likely that other physiological traits not monitored in this study also contribute to c-di-GMP heterogeneity. Alternatively, these cells could represent a transitional state. This single-cell evidence not only complements population-level findings but also highlights the potential role of c-di-GMP in coordinating various virulence traits. Using the fluctuating c-di-GMP levels in individual cells as a cue, the bacteria ensure the division of different virulence features in the population, which is then upheld by positive feedback loops, promoting overall resilience and survival. Our single-cell analysis reveals that c-di-GMP promotes phenotypic diversification by coordinating distinct combinations of virulence factor expression across individual cells, thereby enhancing population-level adaptability.

Since the presence of the T3SS does not influence c-di-GMP levels (Fig. 2A), our study suggests that c-di-GMP determines T3SS activity unidirectionally. Besides c-di-GMP, cAMP acts as a global messenger in bacterial lifestyle transition. In *P. aeruginosa*, both second messengers exert a hierarchical control over pili-mediated surface sensing and biofilm formation[89]. We hypothesize that cAMP initially drives T3SS biogenesis, while c-di-GMP levels, which are linked to cAMP, later influence the secretion behavior of the T3SS. After the surface attachment, elevated c-di-GMP reduces T3SS secretion and ultimately promotes H1-T6SS activity and biofilm formation. Notably, the absence of T3SS further increases biofilm formation, indicating that T3SS could signal a transition to the next stage. This is further supported by an upregulation of H1-T6SS secretion in T3SS-deficient strains (Suppl. Fig. 11). Since the T3SS does not accomplish this by influencing c-di-GMP levels, other mechanisms may be involved.

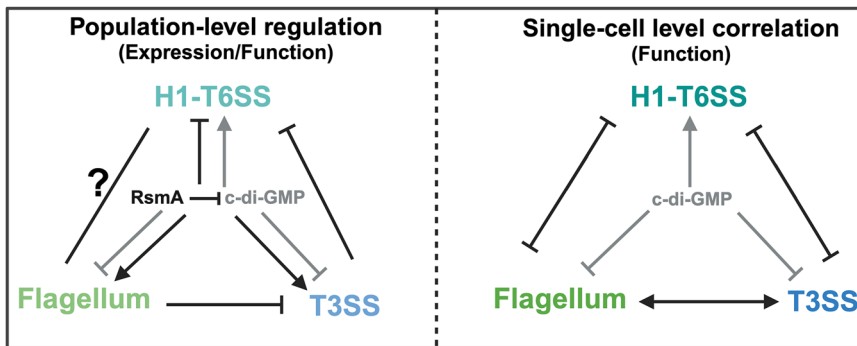

**Fig. 6 | Virulence systems crosstalk in *P. aeruginosa*.** This study focused on H1-T6SS, T3SS and flagellum in *P. aeruginosa*. The population level evidence (left) shows that flagellum existence inhibits T3SS operon transcriptional level[80], and T3SS secretion represses H1-T6SS secretion. The exact relationship between flagellum and H1-T6SS remains unknown. RsmA and c-di-GMP were shown to coordinate the transition among different virulence systems. RsmA decreases c-di-GMP[107]. RsmA (black) promotes flagellum and T3SS expression yet blocks H1-T6SS expression [108,109]. C-di-GMP (gray)

functions in an entirely opposite way to RsmA[57]. The single-cell level evidence (right) shows that flagellum and T3SS function cooperatively in *P. aeruginosa*, while H1-T6SS function is negatively associated with flagellum and T3SS. C-di-GMP is involved in the virulence crosstalk by its positive association with H1-T6SS functional bistability and negative association with T3SS and flagellum functional bistability. Created in BioRender. Chen H. (2025) https://BioRender.com/58n8ik2.

The negative correlation with c-di-GMP that we found for both the flagellum and T3SS indicates a collaborative working pattern of these two machineries during the motile phase, while the positive correlation for H1-T6SS suggests that it is active under different conditions. This virulence crosstalk is best illustrated by the direct correlations among different virulence systems. Our findings show that under conditions activating the T3SS (low [Ca$^{2+}$]), H1-T6SS assembly is significantly reduced in *P. aeruginosa* (Fig. 4A). While calcium has been shown to promote T6SS-mediated killing in *Vibrio fischeri*[90] and to assist in T6SS effector binding to host membranes[91], the T3SS-T6SS relationship has remained less explored. Our single-cell microscopy analysis provides further insight into the crosstalk of these important virulence factors, revealing a negative correlation between T3SS and H1-T6SS at both the expression and assembly level (Fig. 4BC). Similarly, we found the flagella assembled preferably in T3SS-equipped cells, and less in H1-T6SS-positive cells (Fig. 5B, C). Taken together, this highlights a crosstalk strategy leading to a higher number of cells that are specialized for distinct functions than would arise without these connections. Notably, for none of the studied combinations of virulence mechanisms were these correlations exclusive. In vivo infection scenarios, maintaining a range of virulence features could therefore be beneficial for survival, especially when facing fluctuating environments and the ensuing selective pressure, which is in line with a bet-hedging strategy. The correlations between the different virulence systems might further collaboratively promote the fitness of the entire population, while maintaining the least energy consumption through a subtle combination of virulence factors, in line with a division of labor. As an example of this flexible infection strategy, the collaboration between T3SS and the flagellum enhances the efficiency of the early-stage exploitation of the host. Meanwhile, factors such as limited resources and the host's immune response gradually increase the level of c-di-GMP, which alters the population dynamics. This shift allows T6SS-equipped bacteria to become the majority, facilitating a 'hibernation' phase. Incorporating the role of c-di-GMP in virulence regulation and its correlation with the distribution of virulence traits, we propose a regulatory triangle focused on single-cell dynamics (Fig. 6). Considering the role of c-di-GMP in regulating secretion system function and subpopulation formation, we speculate that it acts as a coordination hub of different virulence features, in response to changes in the environment, possibly in combination with RetS.

Interestingly, an earlier study showed that non-flagellated *P. aeruginosa* cells display stronger T3SS-dependent cytotoxicity, suggesting a negative correlation between flagella and T3SS[80]. Surface exploring powered by flagellum motility is important to initiate an infection process through discovering the targets for subsequent attachment. Our data shows that the presence of the T3SS does not influence flagellum-based cell mobility

(Suppl. Fig. 14), which indicates that the presence of the flagellum influences the T3SS unidirectionally. In combination with the collaborative assembly of flagellum and T3SS shown in the microscopy experiments, our analysis lends support to a model where the T3SS can already be assembled alongside the flagellum during surface exploration. As attachment becomes irreversible, flagellar detachment is followed by an increase in T3SS secretion, enhancing host infection for sustained survival benefits. Since c-di-GMP regulates its downstream targets via specific receptors, such as FleQ for flagellum[39], loss of motility can be independent of reduced T3SS secretion.

Taken together, our study integrates the regulation of c-di-GMP, the heterogeneity and crosstalk of virulence systems, and the concept of virulence transition on a single-cell basis. The results illustrate how c-di-GMP and various virulence systems enhance the adaptability of pathogens between different lifestyles. However, the complex cell signaling pathways leave important questions open. Recent discoveries have revealed more about the other two main types of T6SSs in *P. aeruginosa*. The extensive crosstalk observed between H3-T6SS and other virulence factors[87] indicates that this virulence factor, too, is part of a broader regulatory network. Integrating the H2/H3-T6SS and other virulence factors would be required to get a more complete—albeit likely still more complex—picture of the diverse pattern of regulatory states in the pathogenesis of *P. aeruginosa*. Furthermore, investigating the dynamics of the identified subpopulations over extended time frames and in infection models may yield important insights. As a universal second messenger in bacteria, c-di-GMP has a wide range of regulatory roles across various organisms. It will also be interesting to explore whether this virulence transition strategy is present in other Gram-negative species, especially given the diversity of hosts and environments.

Battling bacterial infections remains a complex challenge that demands continuous scientific insight. The bacterial second messenger c-di-GMP has emerged as a key regulator of virulence across diverse microorganisms, though its effects vary between species. For example, in *Salmonella Typhimurium*, the quorum-sensing signal autoinducer-2 (AI-2) can suppress the expression of type III secretion system 1 (T3SS-1) genes by promoting the synthesis of c-di-GMP[92]. Higher c-di-GMP levels have also been associated with stronger biofilm formation and less T3SS-caused macrophage cytotoxicity in *Bordetella bronchiseptica*[93]. Though many individual regulation mechanisms remain unclear, it appears that c-di-GMP plays a universal role in regulating different states of pathogenic bacteria. This makes c-di-GMP an attractive target for therapeutics. Indeed, targeting c-di-GMP levels, for example, with nitric oxide, reduces biofilm levels in cystic fibrosis *P. aeruginosa* isolates[94]. Our findings on the role of c-di-GMP on the activity of three major virulence factors of *P. aeruginosa*, the T3SS, H1-T6SS, and the flagellum, gives valuable insights into how such treatments may be used to

target bacteria across populations, and how the effect on one virulence factor may influence the activity of another, a prerequisite for the application of efficient anti-microbial therapeutics.

## Methods

### Bacterial strains, plasmids, and genetic constructs

A complete list of strains, constructs, and oligonucleotides used in the experiments is listed in Suppl. Table. 2. All *P. aeruginosa* strains are based on the wild-type strain PAO1. Expression vectors and mutator plasmids were transformed into *E. coli* DH5α for cloning, and *E. coli* SM10 λ pir+ for conjugation, respectively. Endogenous fusion proteins used in this study were stably introduced by allelic exchange of the wild-type gene in the *P. aeruginosa* PAO1 genome by two-step homologous recombination for native expression of the proteins of interest[95]. Similarly, strains with gene deletions were generated by homologous recombination and verified by Sanger Sequencing. The Δ*tse6tsi6* deletion strain was labeled fluorescently by expressing sYFP2 from a Tn7 transposon as described by Schlechter and colleagues[96]. For ectopic protein expression in *P. aeruginosa*, the corresponding genes were cloned into pJN105/pUCP20/pMMB67EH-based expression plasmids. These plasmid constructs were sequence-verified (Microsynth Seqlab) before being introduced into the respective bacterial strain via heat-shock transformation[97]. Primers used for constructing the plasmids of this study are listed in Suppl. Table 3.

### Bacterial cultivation

*P. aeruginosa* strains used in this study were routinely grown in Luria-Bertani (LB) medium at 37 °C with shaking at 180 rpm. Overnight cultures were inoculated in triplicates if required. For strains carrying pJN105-based plasmids, Gentamicin (Gm, 40 μg/ml) was also added to ensure plasmid stability. For strains carrying pUCP20 or pMMB67EH-based plasmids, ampicillin (Amp, 300 μg/ml) was also added to ensure plasmid stability. To prepare day cultures, overnight cultures at the stationary phase were inoculated into fresh LB medium with antibiotics (if required) to an $OD_{600}$ of 0.1. To induce ectopic gene expression from the pJN105 plasmid, L-arabinose was also added at a final concentration of 0.2% (w/v). To provide non-secreting or secreting conditions for the T3SS, day cultures were cultivated in non-secreting or secreting medium, LB supplemented with 20 mM $MgCl_2$, 200 mM NaCl, 0.4% (w/v) glycerol and either 5 mM $CaCl_2$ (to inhibit T3SS secretion) or 5 mM EGTA (ethylene glycol-bis(β-aminoethyl ether)-*N*,*N*,*N′*,*N′*-tetraacetic acid, to induce T3SS secretion), respectively.

### Fluorescence microscopy

For fluorescence microscopy, day cultures were cultivated according to the specific experimental conditions and collected after 3 h of growth. Upon completion of the experimental incubation, 600 μl of culture was collected by centrifugation (3 min at 2400 g, room temperature (RT)) and washed once with minimal medium. Samples were then resuspended in 100 μl of minimal medium (100 mM 2-[4-(2-Hydroxyethyl)piperazin-1-yl]ethane-1-sulfonic acid (HEPES) (pH 7.2), 5 mM $(NH_4)_2SO_4$, 100 mM NaCl, 20 mM sodium glutamate, 10 mM $MgCl_2$, 5 mM $K_2SO_4$, 50 mM glycine and 0.2% casamino acids (w/v)). To establish T3SS non-secreting/secreting conditions, 5 mM $CaCl_2$ or 5 mM EGTA was added, respectively. 0.2% (w/v) L-arabinose was added to maintain ectopic gene expression, where required. From this bacterial suspension, 2 μl were spotted onto an agarose pad (1.5% low-melting agarose (Sigma-Aldrich) in minimal medium with required supplements as discussed above) on glass depression slides (Marienfeld), topped with a coverslip (ThermoFisher Scientific). Imaging was conducted using a Deltavision Elite Optical Sectioning Microscope equipped with a UPlanSApo 100×/1.40 oil objective (Olympus) and an EDGE sCMOS_5.5 camera (Photometrics). The GFP/YFP signal was captured using a GFP filter set (excitation: 480/25 nm, emission: 535/28 nm) with an exposure time of 0.3 s, and the mCherry signal was visualized using a mCherry filter set (excitation: 575/25 nm, emission: 625/45 nm) with 0.6 s exposure time. Z-stacks were acquired with 8 slices (Δz = 0.15 μm) per

fluorescence channel. The corresponding images with differential interference contrast channel (DIC) were processed using FIJI (ImageJ 2.1.0). Selected fields of view were identically adjusted for brightness and contrast across compared image sets, and the cell counter macro was used for cells and foci counting. Counting was repeated once for each selected microscopy field to ensure data reliability. For intensity measurements, the background of each selected microscopy field was subtracted individually. The statistical analysis and graph generation were done in GraphPad 10.0 (Dotmatics). For the statistical analysis of the co-existence of two virulence factors, a 2*2 contingency table was used, based on the statistical analysis built into a $\chi^2$ test.

### Statistics and reproducibility

All numerical data values and *p*-values are provided in Suppl. Data 1. The number of biological replicates and details on statistical analyses are provided in the respective figure legends. Error bars represent standard deviations, unless specifically mentioned. Calculations of confidence intervals are based on the t-distribution for smaller sample sizes (n < 30) and normal approximation for larger sample sizes. Effect sizes are Hedges' *g* or Cohen's *d*, for smaller or larger sample sizes, respectively.

### T3SS secretome analysis using shotgun proteomics

The proteomics procedure used for this study is based on a tailored *P. aeruginosa* proteomics analysis method[98]. The respective overnight cultures were inoculated in T3SS non-secreting or secreting medium as described above. After 5 h of growth at 37 °C, $OD_{600}$ was measured, and samples were diluted to the lowest $OD_{600}$ with the same culturing medium. 3 ml of each adjusted culture was centrifuged (15 min at 4000 g, RT) to separate supernatant and cellular fraction, and the supernatant was subsequently filter-sterilized (0.22 μm, Sarstedt) to new tubes. 1.8 ml of supernatant of each strain was transferred to 2 ml Eppendorf tubes pre-filled with 200 μl trichloroacetic acid (TCA) and stored at 4 °C overnight for protein precipitation. All samples were centrifuged (20 min at 21,000 × *g*, 4 °C) and the pellets were washed once with 2 ml cold acetone. The pellets were air-dried at room temperature and stored at −20 °C. Each dried pellet was solubilized in 50 μl 2% (w/v) SLS-ABC buffer (2% sodium lauroyl sarcosinate (SLS), 100 mM ammonium bicarbonate (ABC)) and boiled for 10 min at 99 °C, later sonicated for 30 seconds (Hielscher Ultrasound Technology). Protein reduction was then carried out for 30 min at 37 °C by adding 5 mM Tris (2-carboxyethyl) phosphine (TCEP), followed by alkylation using iodoacetamide (10 mM) for 30 min at 25 °C in the dark. For protein digestion, 1 μg trypsin (Sequencing grade Modified Trypsin, Promega) was added to the sample and incubated at 30 °C overnight. Post digest samples were acidified using 1.5% (v/v) Trifluoroacetic acid (TFA) and C18 purified using Chromabond SpinColumns (Macherey-Nagel). Cartridges were prepared by adding acetonitrile (ACN), followed by equilibration with 0.1% TFA. Peptides were loaded on equilibrated cartridges, washed with 5% ACN and 0.1% TFA containing buffer and finally eluted with 50% ACN and 0.1% TFA.

LC-MS/MS analysis of the peptide samples was carried out on an Exploris 480 instrument connected to an Ultimate 3000 RSLC nano and a nanospray flex ion source (all Thermo Scientific). Peptide separation was performed on a reverse-phase high-performance liquid chromatography column (75 μm × 42 cm) packed in-house with C18 resin (2.4 μm, Dr. Maisch). The peptides were loaded onto a PepMap 100 precolumn (ThermoFisher Scientific) and eluted by a linear ACN gradient from 2 to 25% solvent B over 40 min (solvent A: 0.15% formic acid; solvent B: 99.85% ACN in 0.15% formic acid), and an additional increase of solvent B to 35% over 20 min. The flow rate was set to 300 nl/min. The peptides were analyzed in positive ion mode. The spray voltage was set to 2.3 kV, and the temperature of the heated capillary was set to 275 °C. The data acquisition mode was set to obtain one high-resolution MS scan at a resolution of 60,000 full width at half maximum (at *m/z* 200), followed by MS/MS scans of the most intense ions within 1 s (cycle 1 s). To increase the efficiency of MS/MS attempts, the charged state screening mode was enabled to exclude unassigned and singly

charged ions. The dynamic exclusion duration was set to 14 s. The ion accumulation time was set to 50 ms (MS) and 50 ms at 17,500 resolution (MS/MS). The automatic gain control (AGC) was set to $3 \times 10^6$ for MS survey scan and $2 \times 10^5$ for MS/MS scans. The quadrupole isolation was 1.5 m/z, and collision was induced with an HCD collision energy of 27%.

MS raw data were acquired on an Exploris 480 (Thermo Scientific) in data-independent acquisition (DIA) mode. The funnel RF level was set to 40. For DIA experiments, full MS resolutions were set to 120.000 at m/z 200. AGC target value for fragment spectra was set at 3000%. 45 windows of 14 Da were used with an overlap of 1 Da between m/z 320-950. Resolution was set to 15,000 and IT to 22 ms. Stepped HCD collision energies of 25, 27.5, and 30% were used. MS1 data was acquired in profile, MS2 DIA data in centroid mode.

Analysis of DIA data was performed using the DIA-NN version 1.8[99] using a Uniprot protein database from *Pseudomonas aeruginosa* to generate a data set-specific spectral library for the DIA analysis. The neural network-based DIA-NN suite performed noise interference correction (mass correction, RT prediction and precursor/fragment co-elution correlation) and MS1 peptide precursor signal extraction of the DIA-NN raw data. For the search full tryptic digest was allowed with two missed cleavage sites, and oxidized methionines and carbamidomethylated cysteines. Match between runs and remove likely interferences was active. The precursor FDR was set to 1%. The neural network classifier was set to the single-pass mode, and protein inference was based on genes. Quantification strategy was set to any LC (high accuracy). Cross-run normalization was set to RT-dependent. Library generation was set to smart profiling. DIA-NN outputs were further evaluated using the SafeQuant script modified to process DIA-NN outputs[100,101]. Final graph generation was done via Excel 2016 and GraphPad 10.0.

### Bacterial competition (killing) assay

To compare H1-T6SS-mediated bacteria competition, 0.05 ODu (1 ODu = 1 ml of culture at OD$_{600}$ of 1.0, ~$2 \times 10^8$ *P. aeruginosa* cells, Dong-ju Kim, 2012) of the overnight culture of *P. aeruginosa* prey strain (PAO1 Δ*tse6tsi6* SYFP2, pJN105) and 0.1 ODu of the overnight cultures of the predator strains were mixed together to generate competition pairs. Each pair was centrifuged (3 min at 2400 × $g$, RT) and washed once with fresh LB medium. They were eventually resuspended in fresh LB medium with Gentamicin and L-arabinose, at a total OD$_{600}$ of 0.1. Competition pairs were then added into a 96-well microplate (100 μl/competition pair/per well). Medium-only wells were included for background subtraction. The wells at the perimeter were filled with sterile water, and the lid was sealed to prevent evaporation. Fluorescence was measured every 15 min over 24 h at 37 °C in a Tecan Infinite 2000plate reader. SYFP2 fluorescence (excitation: 500/9 nm, emission: 535/20 nm) for prey survival and absorbance at 600 nm for the growth curve were measured. Data processing and graph generation were done in GraphPad 10.0.

### Acceptor photobleaching fluorescence resonance energy transfer (FRET)

A novel c-di-GMP biosensor was used for the FRET experiments of this study[67]. The functional segments of the biosensor were integrated into pJN105, pMMB67EH and pUCP20 plasmids for expression in *P. aeruginosa*. Overnight cultures of respective strains were inoculated into day cultures (LB medium or secreting medium, if required) supplemented with Gentamicin (40 μg/ml) or Ampicillin (300 μg/ml) and grown at 37 °C for 1.5 h. L-arabinose was added then to induce expression for another 1.5 h. Microscopy slide preparation was the same as described above, except that standard microscopy slides (Carl Roth) were used, and Gene Frame (ThermoFisher Scientific) was applied to seal the samples. Acceptor photobleaching FRET measurements of single cells were performed with a protocol based on refs. 102,103. In this case, a Visitron TIRF/FRAP microscope (Visitron, Puchheim, Germany) equipped with a 2D FRAP Scanner (Galvanometer-controlled 2D module for illumination of single and multiple regions of interest, ROIs) was used. The microscope is based on

a Nikon Ti-E frame, equipped with a perfect focus system (PFS). Images were acquired through a 100X NA 1.49 CFI Apochromat objective using the Visiview (Visitron) software version 5.0. The microscope is connected to a CoolLED pe-4000 light source and an iXon-Ultra-888 EMCCD camera (Andor, Belfast, UK). Images were acquired in multiple channels: mTurquoise2 (FRET donor: excitation 436/20 nm, emission 480/40 nm), YFP (FRET acceptor: excitation 500/40 nm, emission 545/40 nm), mCherry (Fluorescence reporter: Laser widefield excitation 561 nm laser (100 mW), emission 602/45; LP675 nm). For a given fluorescent protein, during the entire study, the pe-4000 output power (p), exposure time (exp) and EM gain (EM) were kept constant for mTurquoise2 to $p = 14\%$, exp = 700 ms, EM = 390; for YFP to $p = 10\%$, exp = 400 ms, EM = 200. In detail, the acceptor photobleaching FRET protocol consisted of the following steps: (i) After acquisition in the mCherry channel, multiple mCherry-positive cells and mCherry-negative ones were manually selected. (ii) One image was acquired in the YFP acceptor channel as a reference for the initial acceptor fluorescence intensity. (iii) Twenty-five "pre-bleach" images were acquired in the mTurquoise 2 donor channel; the relatively high number of frames allows for detrending the data for donor photobleaching[103]. (iv) Acceptor photobleaching was performed, focusing on the multiple selected ROIs, a 515 nm laser (100 mW) set at 60% with a pixel dwell time of 150 ms. To avoid damage to the camera chip, no images were acquired in this step. (v) Twenty-five "post-bleach" images were acquired in the mTurquoise 2 donor channel. (vi) One image was acquired in the YFP acceptor channel to verify the effective photobleaching of the ROIs.

FRET efficiency was calculated in single cells, manually selected in FIJI (ImageJ 2.1.0), as the donor signal increase after acceptor photobleaching divided by the total donor signal after acceptor photobleaching, keeping into account the incomplete photobleaching of the acceptor, according to Equation 1 in Roszik, Szöllősi and Vereb, 2008[104]. Only measurements where the acceptor was effectively bleached (> 75%) were used in the data analysis. To correct for the minor donor photobleaching during steps (b) and (d), we performed linear fitting (RStudio) of the donor fluorescence signal versus time for both pre- and post-acceptor photobleaching 25-frame curves. A final statistical analysis and graph generation were done via GraphPad 10.0.

### c-di-GMP determination via LC-MS/MS

The respective overnight cultures were inoculated in LB medium supplemented with Gentamicin and L-arabinose for c-di-GMP enzyme expression, as described above. The cultures were grown at 37 °C for 3 h and then measured for 0D600. 1 ml of each culture was added into 2 ml Eppendorf tubes pre-filled with 1 ml cold quenching buffer (−80 °C, 70% (v/v) methanol), and mixed well gently. Quenched samples were centrifuged (15 min at 13,000 × $g$, −10 °C) and the supernatant was removed carefully. Extraction buffer (0.5 mM EDTA, 5 mM Trizma base, 50% (v/v) methanol, pH 7.0) and chloroform were added at equal volumes (300 μl/1 ODu cells) to each sample, and vortexed until no pellets were visible. Samples were then placed in the shaker (Eppendorf Thermomixer) for 2 h (0 °C, 600 rpm). Afterward, samples were centrifuged (15 min at 21,000 × $g$, −10 °C) for phase separation. 200 μl of the top-layer supernatant of each sample was filter-transferred (0.2 μm PTFE, Phenomenex) to a new Eppendorf tube. 20 μl of the filtered sample was loaded into 2 ml vial (VWR Avantor) and stored at −20 °C until further analysis. Semi-quantitative determination of cyclic di-GMP was performed using an LC-MS/MS. The chromatographic separation was performed on an Agilent Infinity II 1290 HPLC system using a SeQuant ZIC-pHILIC column (150 × 2.1 mm, 5 μm particle size, peek coated, Merck) connected to a guard column of similar specificity (20 × 2.1 mm, 5 μm particle size, Phenomenex) a constant flow rate of 0.1 ml/min with mobile phase A comprised of 10 mM ammonium acetate in water, pH 9, supplemented with medronic acid to a final concentration of 5 μM and mobile phase B comprised of 10 mM ammonium acetate in 90:10 acetonitrile to water, pH 9, supplemented with medronic acid to a final concentration of 5 μM at 40 °C. The injection volume was 5 μl. The mobile phase profile consisted of the following steps and linear gradients: 0–1 min

constant at 75% B; 1–6 min from 75 to 40% B; 6 to 9 min constant at 40% B; 9–9.1 min from 40 to 75% B; 9.1 to 20 min constant at 75% B. An Agilent 6495 ion funnel mass spectrometer was used in negative ionization mode with an electrospray ionization source and the following conditions: ESI spray voltage 3000 V, nozzle voltage 1000 V, sheath gas 400 °C at 11 l/min, nebulizer pressure 20 psig and drying gas 100 °C at 11 l/min. The compound was identified based on its mass transition and retention time compared to a chemically pure standard. Chromatograms were integrated using MassHunter software (Agilent, Santa Clara, CA, USA). Metabolite abundance was determined based on peak areas. Mass transitions, collision energies, Cell accelerator voltages, and Dwell times have been optimized using a chemically pure standard. The parameter settings are listed in Suppl. Table 4.

### Biofilm staining
Biofilm formation was examined during growth in a 96-well plate, using crystal-violet staining as described previously[105]. In brief, overnight cultures were diluted to $OD_{600}$ of 0.05 into fresh LB medium or secreting medium, supplemented with Gentamicin and L-arabinose if required. Diluted samples were added to the 96-well microplate (100 μl/strain/per well). The wells at the perimeter were filled with sterile water, and the lid was sealed to prevent evaporation. Incubation was carried out at 37 °C for 24-h with shaking at 180 rpm. After the incubation, the supernatant was carefully removed, and the microplate was rinsed twice in distilled water to further remove the debris. The biofilm biomass attached to each well surface was stained by 150 μl crystal violet (0.1% (w/v)) staining for 15 min. Excess crystal violet was carefully rinsed off with distilled water twice, and then the microplate was left at room temperature for drying overnight before taking digital photographs. After the image recording, 150 μl of 30% acetic acid was added per well with shaking for 15 min to dissolve the crystal violet. 100 μl of the dissolved solution per well was transferred to a new microplate and sent to the plate reader for quantification. Buffer-only wells were included for background subtraction. Crystal violet was read at absorbance 550 nm. Data processing and graph generation were done using GraphPad 10.0.

### Flagellum swimming motility assay
Flagellum swimming motility was examined as previously published[106]. Overnight cultures were inoculated into day cultures (non-secreting or secreting medium) and incubated at 37 °C for 3 h. After that, respective cultures were toothpick-inoculated into swimming plates (secreting and non-secreting medium supplied with 0.3% (w/v) agar) and incubated at 37 °C for 24 h. Digital photographs were taken, and the diameters of the swim zones were measured in millimeters.

### Western blotting
For protein expression and stability tests, bacteria were cultured according to the experimental conditions described above. 2 ml of culture was collected and centrifuged (3 min at $2400 \times g$, RT) to remove the supernatant. The cell pellets were resuspended in 1% (w/v) SDS buffer with the volume calculated to accommodate the SDS–PAGE loading concentration at 0.3 ODu/15 μl. Resuspended samples were boiled for 10 min at 99 °C, and then sonicated for 30 seconds. The processed samples were loaded identically into 2 SDS–PAGE gels with 15 μl/well. Proteins were separated on 15% (w/v) SDS–PAGE gels with Precision Plus All Blue Prestained (BioRad) as a size standard. One SDS–PAGE gel was stained with FastGene-Q-stain (NipponGenetics) for visualization, and the counterpart was transferred onto a nitrocellulose membrane with BioRad semi-dry transfer system (1.3 A; 25 V; 20 min) for immunoblotting. Membranes were blocked with non-fat milk (5% (w/v) in PBS) and rinsed with PBS-T (0.2% (v/v) Triton X-100 in PBS) for subsequent antibody attachment. Primary rabbit antibodies against mCherry (Biovision 5993, 1:2000) were used in combination with secondary anti-rabbit antibodies conjugated to horseradish peroxidase (HRP) (Sigma, A8275, 1:5000). Primary mouse antibodies against GFP (Proteintech 66002-1, 1:1000) were used in combination with secondary anti-mouse antibodies conjugated to horseradish peroxidase (HRP) (GE Healthcare NXA931,

1:5,000). For the detection of chemiluminescence signals, ECL chemiluminescence substrate (Millipore, WBLUF0500) was used in a LAS-4000 Luminescence imager (GE Healthcare).

## Data availability
All data supporting the findings of this study are available within the paper and its Supplementary Information. Uncropped and unedited blot images are provided in Suppl. Fig. 15. All numerical data values and *p*-values are provided in Suppl. Data 1. All other data are available from the corresponding author on reasonable request.

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

## Acknowledgements
Work in the Diepold lab was supported by the Max Planck Society. Work in the Unterweger group was supported by the German Federal Ministry for Education and Research (grant 01KI2020) and the Deutsche Forschungsgemeinschaft (RU5042, EXC2167, CRC1182).

## Author contributions
H.C. performed the majority of measurements and data processing and participated in writing and revising the manuscript. G.M. and L.W. contributed the c-di-GMP reporter and helped with FRET data acquisition and processing. O.V. and M.R. provided *P. aeruginosa* strains for the T6SS analysis. T.G. performed the proteomics analysis. N.P. performed the metabolomics analysis. D.U. and V.S. provided strains and participated in data interpretation and writing the manuscript. A.D. provided the study concept, supervision, and participated in data analysis and writing and revision of the manuscript.

## Funding

## Competing interests
The authors declare no competing interests.
