## [Transparent Peer Review file · Communications Biology]

Coordination of virulence factors and lifestyle transition in *Pseudomonas aeruginosa* through single-cell analysis

Corresponding Author: Professor Andreas Diepold

Version 0:

Reviewer comments:

Reviewer #1

(Remarks to the Author)

The paper "Coordination of virulence factors and lifestyle transition in *Pseudomonas aeruginosa* through single-cell analysis" uses sophisticated single-cell analysis to investigate how c-di-GMP controls the virulence systems (T3SS, H1-T6SS, and flagellum). Strong methodological rigor, lucid presentation, and possible therapeutic implications make it a valuable resource for understanding bacterial heterogeneity and lifestyle shifts.

There are still some minor and major revisions need to go over to guarantee a full and accurate evaluation, which I will address in my assessment.

Minor revisions.

- 1- The authors have to choose the keywords from MeSH/NLM.
- 2- Please provide a recent reference preferably from 2024 that confirmed that *Pseudomonas aeruginosa* remains classified as a "high priority" pathogen by WHO. Incorporating the most recent categorization will enhance the manuscript's correctness and relevancy, as WHO regularly changes its priority pathogen list.
- 3- Enhance statistical analysis by incorporating effect sizes/confidence intervals.
- 4- Improve figure/table clarity with better labeling and annotations.
 - 3.a- Figure 1: Y-axis labels in panels B and C should be more descriptive.
 - 3.b- Figure 4: The microscopy pictures in Panel B require scale bars and clearer annotations.
 - 3.c- Supplementary figures: Some require improved labels and legends for accessibility.
- 5- Comparisons to other bacterial pathogens that control virulence through c-di-GMP should be part of the discussion.
- 6- Discuss the study's shortcomings, such as population dynamics and the role of other T6SS.

Major revisions.

- 1- Provide additional mechanistic insights into how c-di-GMP modulates virulence.
- 2- Discuss the biological significance of the findings in in vivo infection scenarios.
- 3- Expand on the broad implications for bacterial pathogenesis and therapeutic methods.

Reviewer #2

(Remarks to the Author)

The authors used single-cell studies to study the effects of high and low c-di-GMP levels in *P. aeruginosa*'s T6SS and T3SS systems. I have several major comments:

- [1] The authors artificially increase or decrease c-di-GMP levels via the use of YhjH and DgcA, but both genes may have other potential unrelated effects on the bacterial cells. It is important to confirm the observations in physiologically relevant conditions such as biofilm formation and biofilms dispersal.
- [2] The findings that increased c-di-GMP levels increase T6SS and lower T3SS are well-studied many years ago (<https://pubmed.ncbi.nlm.nih.gov/21955777/>), so the authors have to show new insights based on the single-cell sequencing.

[3] The authors also used the FRET system which was also previously published, so a novel c-di-GMP quantification system should be used for observation.

[4] The authors focused on bacterial killing by T6SS, but seems to neglect the T3SS. It is important to show that T3SS-mediated killing is reduced by increase in c-di-GMP signaling. Furthermore, other studies had shown that decrease in c-di-GMP can induce T3SS (<https://www.nature.com/articles/ncomms5462>), so it is important to put into context those studies and cite them.

[5] Fig 7 showed additional information which were not shown in the study, such as acute and chronic infection, so please remove them. Moreover, this information on the mechanisms was similar to Fig 6, so it is advised to merge them together.

[6] Supp Fig. 5 should show wild-type data, instead of being processed as a ratio.

[7] Supp Fig. 7 should show individual channels, instead of just a merged image, as it is difficult to discern. This applies to other images in other figures too.

Reviewer #3

(Remarks to the Author)

In this work, the authors investigate the expression of the flagellum, the type III secretion system (T3SS) and the H1-type VI secretion system (H1-T6SS, one of three T6SS in the PAO1 strain) on the single cell level in *Pseudomonas aeruginosa* PAO1 in response to different levels of cyclic di-GMP, a ubiquitous second messenger directing the sessile versus motile life style in bacteria. For this purpose, they constructed (translational?) fusions of cytosolic structural proteins, fluorescent reporter fusions to promoters and a cyclic di-GMP responsive reporter. As expected and previously shown on the population level, they found that while T3SS and the flagellum are functionally co-regulated, T6SS is counter-regulated on different levels.

General comments

This work has prepared some very useful tools to study expression of major cyclic di-GMP affected nanomachines on the single cell level. What is missing is the introduction, construction scheme, genomic context and demonstrated validation of the reporter systems at the beginning of the description of the work in the manuscript (associated with the first figure and/or successively when relevant).

As there is an accurate estimation of the fraction/overlap of the cell population producing a nanomachine(s), can there be any drawback by using a heterologous diguanylate cyclase and phosphodiesterase? Does the fraction of cells/overlap significantly shift by using another diguanylate cyclase/phosphodiesterase? Is it known which DGC/PDE is dedicated to regulation the expression of the T2SS and T6SS in *P. aeruginosa*?

In addition, figure legends are often missing accurate descriptions and explanations that enable a comprehensive understanding of the figures without consultation of result description and/or material and methods.

Also, while quite popular as a justification, I miss convincing arguments/data in the work that indicate potential new targets for pharmaceutical intervention.

Further, although recent references show the advancements in the field, references to several original works reporting fundamental concepts and discoveries should be added.

Other comments (not necessarily comprehensive)

l. 62: carbapenem resistant *Pseudomonas aeruginosa* are critical priority.

l. 66 ff: This description of T3SS functionality is blurry.

l. 103: What about the Psl exopolysaccharide?

l. 107: not sure that motility is required for initial dispersal, is there a reference?

l.110: Has this been shown in this way?

l. 148: What means 'indicates direct connections between T3SS, H1-T6SS and flagellum'?

l. 169: substrate?, are these the effector or chaperon-effector complexes?

l. 522 and elsewhere: gentamicin

Figure 1 and elsewhere: Representative micrographs and Western blots should be shown here or in the supplementary data, not only the numbers.

Figure 3 and elsewhere: Preferable also micrographs with lower magnification with more cells should be shown.

Figure S5: data from different experimental conditions cannot be placed in the same graph. In addition, as the figure clearly shows that wild type biofilm formation is substantially different, this is misleading. Also the data showing the other deletion mutant under the same conditions is missing.

Figure S3: the micrograph is identical to the one in Figure 3, why?

Figure S8: An additional documentation with lower magnification showing more cells would be even more convincing.

Figure S10: were taken from 15 different fields in 3 different experiments, would be informative to indicate number of investigated cells in each case

Figure S11/12: This refers well to the swimming speed?

Version 1:

Reviewer comments:

Reviewer #1

(Remarks to the Author)

The authors have addressed all the comments and suggestions raised in the previous round. I recommend the manuscript. Just a few corrections.

In line 559, please refrain from mentioning the full name of the second messenger cyclic di-GMP (c-di-GMP) again; it has already been mentioned before.

In line 560, make *Salmonella Typhimurium italic*,

Reviewer #2

(Remarks to the Author)

The authors had addressed most of my comments, except for Comment 1. I am not convinced about the use of foreign plasmids with the DGC and PDE, without control experiments to back the authors' claim that both plasmids have no effect on bacterial physiology. The appropriate controls to use are the bacterial cells with empty plasmids without the DGC and PDE. A growth curve or transcriptomics experiments will be appropriate to compare the bacteria with no plasmids and bacteria with empty plasmids.

Reviewer #3

(Remarks to the Author)

I have no further comments.

Reviewers' comments:

Reviewer #1 (Remarks to the Author):

The paper "Coordination of virulence factors and lifestyle transition in *Pseudomonas aeruginosa* through single-cell analysis" uses sophisticated single-cell analysis to investigate how c-di-GMP controls the virulence systems (T3SS, H1-T6SS, and flagellum). Strong methodological rigor, lucid presentation, and possible therapeutic implications make it a valuable resource for understanding bacterial heterogeneity and lifestyle shifts.

There are still some minor and major revisions need to go over to guarantee a full and accurate evaluation, which I will address in my assessment.

We thank the reviewer for this positive assessment!

Major revisions.

1- Provide additional mechanistic insights into how c-di-GMP modulates virulence.

As the reviewer states in his/her summary, the focus of this work is on the effect of c-di-GMP on the T3SS, H1-T6SS, and the flagellum. While mechanistic insights, such as the identification and characterization of c-di-GMP-binding proteins directly influencing the expression or activity of the systems, are an important direction of future research by our and other groups, they were not the focus of this study, which described the c-di-GMP influence on the secretion systems and their interplay on the level of individual cells. We more clearly highlight this focus, and the limitations of the study, in the discussion (e.g. lines 549-552, all line numbers refer to the version of the manuscript in which the changes are marked).

2- Discuss the biological significance of the findings in in vivo infection scenarios.

We thank the reviewer for pointing out that this aspect was not adequately covered so far. We have now added an additional paragraph discussing this aspect (lines 558-564) in addition to the parts of the discussion that had already dealt with this point in the original manuscript (e.g. lines 158-166, 510-513).

3- Expand on the broad implications for bacterial pathogenesis and therapeutic methods.

While we are careful to not overstate the direct implications of our basic research on therapeutic methods, we have added a paragraph discussing these aspects in the revised manuscript (lines 564-570).

Minor revisions.

1- The authors have to choose the keywords from MeSH/NLM.

We have adapted the keywords, which are now all included in MeSH.

2- Please provide a recent reference preferably from 2024 that confirmed that *Pseudomonas aeruginosa* remains classified as a "high priority" pathogen by WHO. Incorporating the most recent categorization will enhance the manuscript's correctness and relevancy, as WHO regularly changes its priority pathogen list.

We now include this reference.

3- Enhance statistical analysis by incorporating effect sizes/confidence intervals.

We now provide the confidence intervals for the numerical analyses (Fig. 1BCE, 2AB, 3AB, 4B, 5A, Suppl. Fig. 1AB, 5, 7, 9, 11, 12), based on the t-distribution for smaller sample sizes ($n < 30$) and normal approximation

for larger sample sizes. We have taken this chance to display all numerical data in an Excel file, which is provided as a Supplementary file. This file also contains the confidence intervals and the effect sizes (Hedges' g or Cohen's d , for smaller / larger samples sizes, respectively) for Fig. 1B, 2AB, 3AB, 4B, 5A, S1B, 7, 9. In the cases where we considered this to be most relevant, we also mention the effect size and/or confidence intervals in the main text or figure legends.

4-Improve figure/table clarity with better labeling and annotations

3.a- Figure 1: Y-axis labels in panels B and C should be more descriptive.

We assume that this comments refers to panels B and D, where we changed the labels to indicate the nature of the detected foci (which we also include in the y axis label of panel A). For panel C, "protein intensity" is the common assignment in the field. We now include the additional information that this is the measured signal strength that reflects the abundance of a protein in a sample in the figure legend.

3.b- Figure 4: The microscopy pictures in Panel B require scale bars and clearer annotations.

We have changed the figure annotations, which now clearly indicate the nature of the detected proteins. All micrographs in the figure include a scale bar.

3.c- Supplementary figures: Some require improved labels and legends for accessibility.

We have made changes to labels and legends of several supplementary figures, including Fig. S3 (now S4), S5 (now S6), S9 (now S11).

5- Comparisons to other bacterial pathogens that control virulence through c-di-GMP should be part of the discussion.

The discussion now includes an additional final paragraph (lines 558-570), which discusses the role of c-di-GMP across different bacteria and the resulting options for therapeutic intervention.

6- Discuss the study's shortcomings, such as population dynamics and the role of other T6SS.

Both of these points are now more clearly stated in lines 549-552.

Reviewer #2 (Remarks to the Author):

The authors used single-cell studies to study the effects of high and low c-di-GMP levels in *P. aeruginosa*'s T6SS and T3SS systems. I have several major comments:

[1] The authors artificially increase or decrease c-di-GMP levels via the use of YhjH and DgcA, but both genes may have other potential unrelated effects on the bacterial cells. It is important to confirm the observations in physiologically relevant conditions such as biofilm formation and biofilms dispersal.

The use of heterologous and catalytically well-characterized c-di-GMP-modulating enzymes such as YhjH (a phosphodiesterase with a conserved EAL domain) and DgcA (a diguanylate cyclase with a GGDEF domain) is a widely accepted and advantageous strategy for manipulating intracellular c-di-GMP levels.

Indeed, this approach has been successfully applied in multiple systems, including *Pseudomonas aeruginosa*, where heterologous expression of YhjH was used to study biofilm dispersal (Christensen et al., 2013), and where DgcA from *Mycobacterium leprae* was shown to be functional and to alter c-di-GMP-related phenotypes (Rotcheewaphan et al., 2016). In our study, we observed no unrelated or adverse cellular effects beyond the expected modulation of c-di-GMP-associated phenotypes. Importantly, in addition to the expected results for T6SS competition (which is increased upon YhjH expression and decreased upon DgcA expression, Fig. 1E), we also demonstrate a clear link between elevated c-di-GMP levels (via DgcA expression) and enhanced biofilm formation, as shown in Supplementary Figure 5B.

The main reason for choosing this heterologous enzyme was that, because these enzymes are not native to the studied organism and contain no major additional domains, they are less likely to interact with host regulatory networks beyond their enzymatic activity on c-di-GMP, reducing confounding effects associated with native regulators. Our results support the specificity of our approach and confirm that the observed phenotypes are indeed due to altered c-di-GMP levels. We now outline this reasoning in lines 174-176 (all line numbers refer to the version of the manuscript in which the changes are marked).

[2] The findings that increased c-di-GMP levels increase T6SS and lower T3SS are well-studied many years ago (<https://pubmed.ncbi.nlm.nih.gov/21955777/>), so the authors have to show new insights based on the single-cell sequencing.

We agree with the reviewer that the reciprocal regulation of T6SS and T3SS by c-di-GMP has been previously described (e.g., Moscoso et al., 2011). The novelty of our study lies in the single-cell resolution of these dynamics, which allows us to uncover population-level heterogeneity that cannot be detected in bulk measurements. Specifically, our data reveal that c-di-GMP does not simply act as a binary switch between T3SS and T6SS but instead shapes the emergence of distinct subpopulations with defined combinations of virulence traits. This heterogeneity likely reflects adaptive strategies for resilience in fluctuating microenvironments.

Importantly, our approach enables direct correlation of T3SS and T6SS expression within individual cells, allowing us to observe that the activation of these systems is not merely mutually exclusive but part of a broader landscape of virulence combinations. This underscores a coordinating role of c-di-GMP in fine-tuning the phenotypic diversification of the population. While the manuscript already addressed these aspects conceptually (e.g., at the end of the introduction), we now highlight them more explicitly in the revised version, e.g. with added statements at lines 447-449 and 457-461, to clarify the novelty of our single-cell approach and its contribution to understanding population heterogeneity and virulence coordination.

[3] The authors also used the FRET system which was also previously published, so a novel c-di-GMP quantification system should be used for observation.

We acknowledge that the FRET-based c-di-GMP reporter system is not a newly developed technique. We deliberately chose this established and well-characterized method, as it offers specific advantages that are particularly well-suited to the goals of our study. Rather than developing a novel and untested quantification system, we adapted and optimized the FRET sensor for a new application: correlating intracellular c-di-GMP levels with virulence factor expression at the single-cell level—an approach that, to our knowledge, has not been previously reported.

While technically more demanding than traditional reporters such as LacZ or EGFP, the FRET system enables (1) real-time, non-destructive quantification of c-di-GMP, (2) single-cell resolution without the need for an external normalization fluorophore, and (3) in our case, the use of a recently improved variant with enhanced affinity and dynamic range (Wang et al., 2024). We now clarify these points and the rationale for using this sensor more explicitly in the revised manuscript at lines 248-254.

[4] The authors focused on bacterial killing by T6SS, but seems to neglect the T3SS. It is important to show that T3SS-mediated killing is reduced by increase in c-di-GMP signaling. Furthermore, other studies had shown that decrease in c-di-GMP can induce T3SS (<https://www.nature.com/articles/ncomms5462>), so it is important to put into context those studies and cite them.

While, as the reviewer notes, host cell killing is a key outcome of T3SS activity, we intentionally analyzed T3SS at the level of effector secretion, which provides a direct and quantitative measure of its function. The correlation between effector secretion and cytotoxic effects has been established in various settings (e.g., doi: 10.1128/CMR.00013-07; doi: 10.3390/biom11020316). Compared to phenotypic assays such as host cell killing—which can be affected by confounding factors like bacterial adhesion, replication, or host

susceptibility—secretion profiling offers clearer insight into T3SS regulation. It also avoids technical challenges such as maintaining precise c-di-GMP induction during extended killing assays. Thus, secretion analysis provides a robust and specific readout of T3SS activity under different c-di-GMP conditions.

We appreciate the reviewer's reference to the study (Chua et al., 2014), which provides valuable context regarding the induction of T3SS under low c-di-GMP conditions. We now cite this manuscript, along with a short clarification of the rationale of our experiments, in lines 193-194 and 197-199 of the revised manuscript.

[5] Fig 7 showed additional information which were not shown in the study, such as acute and chronic infection, so please remove them. Moreover, this information on the mechanisms was similar to Fig 6, so it is advised to merge them together.

Fig. 6 aims to provide a concise summary of our results by illustrating the correlations among the virulence systems analyzed in this study. Fig. 7 was intended to go a step further and highlight the broader relevance of our findings in the context of real infection scenarios. However, we agree that it extends beyond our direct experimental data and is partially redundant to Fig. 6. We have therefore removed Fig. 7, as requested.

[6] Supp Fig. 5 should show wild-type data, instead of being processed as a ratio.

In response to this suggestion and a comment by reviewer 3, we split the figure into two parts and directly display the individual OD550 values of the Crystal Violet assay.

[7] Supp Fig. 7 should show individual channels, instead of just a merged image, as it is difficult to discern. This applies to other images in other figures too.

As suggested, we added the separate channels in Fig. S7 (now Fig. S8), as well as Fig. 4. We left the merged images in the cases where the signals are clearly discernible in larger fields of view, such as Fig. 5.

Reviewer #3 (Remarks to the Author):

In this work, the authors investigate the expression of the flagellum, the type III secretion system (T3SS) and the H1-type VI secretion system (H1-T6SS, one of three T6SS in the PAO1 strain) on the single cell level in *Pseudomonas aeruginosa* PAO1 in response to different levels of cyclic di-GMP, a ubiquitous second messenger directing the sessile versus motile life style in bacteria. For this purpose, they constructed (translational?) fusions of cytosolic structural proteins, fluorescent reporter fusions to promoters and a cyclic di-GMP responsive reporter. As expected and previously shown on the population level, they found that while T3SS and the flagellum are functionally co-regulated, T6SS is counter-regulated on different levels.

We thank the reviewer for the positive and constructive feedback. As mentioned by the reviewer, our work confirms a series on previous reports on the influence of c-di-GMP on the individual virulence factors, and adds the analysis on the single-cell level, allowing to distinguish subpopulations. Our responses to the individual points are listed below.

General comments

This work has prepared some very useful tools to study expression of major cyclic di-GMP affected nanomachines on the single cell level. What is missing is the introduction, construction scheme, genomic context and demonstrated validation of the reporter systems at the beginning of the description of the work in the manuscript (associated with the first figure and/or successively when relevant).

The FRET biosensor that we used for this study has been developed by the group of Victor Sourjik. Its design and validation in *E. coli* is described in a preprint, which we reference. In our manuscript, we validated the reporter system in *P. aeruginosa* and confirmed its response to c-di-GMP level changes (Suppl. Fig. 1B). To indicate this more clearly, we now briefly introduce the system and the reasons for the choice of this reporter in the last paragraph of the introduction (Line 156-158, all line numbers refer to the version of the manuscript in which the changes are marked), in addition to the previous description of the biosensor including the reference prior to its application in Figure 2, where it was first used.

As there is an accurate estimation of the fraction/overlap of the cell population producing a nanomachine(s), can there be any drawback by using a heterologous diguanylate cyclase and phosphodiesterase? Does the fraction of cells/overlap significantly shift by using another diguanylate cyclase/phosphodiesterase? Is it known which DGC/PDE is dedicated to regulation the expression of the T3SS and T6SS in *P. aeruginosa*?

In *P. aeruginosa*, the native c-di-GMP-metabolizing enzymes WspR and PA2133 have been shown to play important roles in regulating T3SS and T6SS expression (e.g., Moscoso et al., 2011). However, as native diguanylate cyclases (DGCs) and phosphodiesterases (PDEs) often participate in multiple regulatory circuits and may contain additional signaling domains, their overexpression or manipulation can result in pleiotropic effects beyond c-di-GMP modulation. To avoid such confounding influences, we deliberately chose heterologous and catalytically well-characterized enzymes—YhjH (a PDE with a conserved EAL domain) and DgcA (a DGC with a GGDEF domain)—which lack additional regulatory domains and are not native to *P. aeruginosa*. This strategy is widely accepted and allows specific modulation of c-di-GMP levels without undesired interference with endogenous regulatory pathways. While we did not directly compare the effects of multiple DGCs/PDEs on subpopulation distributions in this study, the fact that expression of heterologous enzymes alone is sufficient to shift the fraction of nanomachine-producing cells supports the idea that c-di-GMP acts as a general and global regulator of virulence pathway balance, rather than being tied exclusively to the function of specific native enzymes. We now clarify these points more explicitly in the revised text at lines 174-176.

In addition, figure legends are often missing accurate descriptions and explanations that enable a comprehensive understanding of the figures without consultation of result description and/or material and methods.

We thank the reviewer for the comment, and revised the majority of the figure legends to allow for a comprehensive understanding of the figure. In the few cases where this would have resulted in an overly long figure legend, we specifically refer to the main text and material and methods.

Also, while quite popular as a justification, I miss convincing arguments/data in the work that indicate potential new targets for pharmaceutical intervention.

We appreciate the reviewer's comment. As a general principle, effective therapeutic targeting requires a thorough understanding of the regulatory systems involved. Our study contributes to this by dissecting how c-di-GMP levels shape not only individual virulence traits, but also the overall phenotypic distribution within the population. This reinforces the notion that c-di-GMP acts as a global coordinator of lifestyle and virulence decisions. While the specific molecular links between c-di-GMP and individual effectors remain incompletely understood, our data support the idea that c-di-GMP represents a universal regulatory hub across multiple pathogenic strategies. This strengthens its candidacy as a therapeutic target, especially when direct targeting of specific effectors is not feasible.

We now explicitly address this point in the final paragraph of the discussion, where we discuss the role of c-di-GMP in other pathogens, refer to studies showing the effect of c-di-GMP manipulations on biofilm dispersal, and highlight the potential for intervention, as manipulating c-di-GMP also can steer bacterial

populations into phenotypic states more vulnerable to immune clearance or antibiotic action (lines 558-570).

Further, although recent references show the advancements in the field, references to several original works reporting fundamental concepts and discoveries should be added.

We thank the reviewer for this suggestion. We have now included references to foundational studies that established the role of c-di-GMP in bacterial signaling, motility, and biofilm formation, including Ross et al. (1987), Hickman and Harwood (2005), and Kulasekara et al. (2006), and classical reviews introducing concepts, such as by Jenal and Malone (2006) and Regine Hengge (2009). We also include references to the works by Christen et al. (2010) and Valentini and Filloux (2016) and c-di-GMP variation between cells and the role of phenotypic heterogeneity. Together these works provide essential context for the regulatory pathways discussed in our manuscript. The revised paragraph can be found in lines 152-158.

Other comments (not necessarily comprehensive)

I. 62: carbapenem resistant *Pseudomonas aeruginosa* are critical priority.

We now include this important specification.

I. 66 ff: This description of T3SS functionality is blurry.

We have changed the description (lines 67-69).

I. 103: What about the Psl exopolysaccharide?

We now mention Psl and its role as an extracellular signal to stimulate c-di-GMP production at this point (lines 109-111).

I. 107: not sure that motility is required for initial dispersal, is there a reference?

In *P. putida*, the PDE DipA (dispersion induced phosphodiesterase A) was found to increase swarming motility and reduce bacteria attachment and polysaccharide production by decreasing c-di-GMP levels (Roy et al., 2012). We have added this reference; additional links between c-di-GMP modifying enzymes and dispersal are listed in the referenced review by Ma et al. (2020).

I.110: Has this been shown in this way?

The role of c-di-GMP in mediating the motile/sessile switch is well established, and both biofilm formation and motility are key phenotypes contributing to environmental adaptation and survival in *P. aeruginosa*. High intracellular c-di-GMP levels promote biofilm formation, which is associated with increased tolerance to antibiotics and immune evasion in chronic infections. In contrast, low c-di-GMP levels promote flagellum-mediated motility, aiding in surface colonization and dispersal in fluctuating environments. These opposing phenotypes, regulated by c-di-GMP, support bacterial persistence under varying conditions and thereby constitute an adaptive strategy. We have reworded the sentence as follows to hopefully clarify our argumentation: "This c-di-GMP-mediated cycle switch between motility and biofilm formation enables *P. aeruginosa* to dynamically adapt to changing environments by promoting dispersal under favorable conditions and persistence under stress, thereby promoting its survival."

I. 148: What means 'indicates direct connections between T3SS, H1-T6SS and flagellum'?

We thank the reviewer for pointing out this vague statement, which we changed to "Single-cell microscopy further revealed correlations between the presence of the T3SS, H1-T6SS, and the flagellum."

I. 169: substrate?, are these the effector or chaperon-effector complexes?

Changed to “effector recognition”.

I. 522 and elsewhere: gentamicin

Thanks for pointing out this spelling error. We corrected it throughout.

Figure 1 and elsewhere: Representative micrographs and Western blots should be shown here or in the supplementary data, not only the numbers.

In general, quantifying the microscopy images is the best practice to investigate distributions such as the proportions of T3SS- and T6SS-positive cells and their correlation with c-di-GMP, and representative images can be misleading. Having said this, we now provide images for illustration purposes for Fig. 1B in a new Supplementary Figure (Suppl. Fig. 3). Representative Western blots are shown in Suppl. Fig. 2.

Figure 3 and elsewhere: Preferable also micrographs with lower magnification with more cells should be shown.

The micrographs in the figure are provided to illustrate the distinction between T3SS/T6SS-positive and -negative cells, for the analysis. Nevertheless, we now include larger fields of view for these micrographs, as well as for the flagellum (FlIM-Cherry), used in Fig. 5, in a new Supplementary Figure (Suppl. Fig. 9).

Figure S5: data from different experimental conditions cannot be placed in the same graph. In addition, as the figure clearly shows that wild type biofilm formation is substantially different, this is misleading. Also the data showing the other deletion mutant under the same conditions is missing.

As the reviewer correctly states and as mentioned in the manuscript, the conditions used for T3SS analysis and T6SS analysis differ, which also can have an effect on phenotypes that are not directly related. We agree that joining these conditions in a single figure was incorrect and have now separated the figure into two subpanels, as was also requested by another reviewer.

Figure S3: the micrograph is identical to the one in Figure 3, why?

This comment most likely refers to Fig. S7 (now Fig. S8), where we show the good correlation between T3SS assembly and activity, as measured by EGFP-SctQ spot formation and pExoS-mCherry reporter activity, respectively. As this information is not needed for the supporting micrograph in Fig. 3A, we have replaced the micrograph in Fig. 3 by another image of the mCherry channel alone, which also brings the image in line with the image for Fig. 3B.

Figure S8: An additional documentation with lower magnification showing more cells would be even more convincing.

As requested, we show an additional figure with lower magnification in the revised version.

Figure S10: were taken from 15 different fields in 3 different experiments, would be informative to indicate number of investigated cells in each case

We now include the range of numbers for each field of view, in addition to the exact total number of cells.

Figure S11/12: This refers well to the swimming speed?

While the swimming diameter is a well-established and widely used indicator of swimming capability, we did not measure the speed of individual bacteria under the microscope. We chose swimming diameter because it provides a robust, standardized, and reproducible measure that allows for high-throughput comparison between conditions. This rationale has now been added to the figure legend of S11 (now S13).

Dear Editor,

The paper "**Coordination of virulence factors and lifestyle transition in *Pseudomonas aeruginosa* through single-cell analysis**" uses sophisticated single-cell analysis to investigate how c-di-GMP controls the virulence systems (T3SS, H1-T6SS, and flagellum). Strong methodological rigor, lucid presentation, and possible therapeutic implications make it a valuable resource for understanding bacterial heterogeneity and lifestyle shifts.

There are still some minor and major revisions need to go over to guarantee a full and accurate evaluation, which I will address in my assessment.

Minor revisions.

- 1- The authors have to choose the keywords from MeSH/NLM.
- 2- Please provide a recent reference preferably from 2024 that confirmed that *Pseudomonas aeruginosa* remains classified as a "high priority" pathogen by WHO. Incorporating the most recent categorization will enhance the manuscript's correctness and relevancy, as WHO regularly changes its priority pathogen list.
- 3- Enhance statistical analysis by incorporating effect sizes/confidence intervals.
- 4- Improve figure/table clarity with better labeling and annotations.
 - 3.a- Figure 1: Y-axis labels in panels B and C should be more descriptive.
 - 3.b- Figure 4: The microscopy pictures in Panel B require scale bars and clearer annotations.
 - 3.c- Supplementary figures: Some require improved labels and legends for accessibility.
- 5- Comparisons to other bacterial pathogens that control virulence through c-di-GMP should be part of the discussion.
- 6- Discuss the study's shortcomings, such as population dynamics and the role of other T6SS.

Major revisions.

- 1- Provide additional mechanistic insights into how c-di-GMP modulates virulence.
- 2- Discuss the biological significance of the findings in in vivo infection scenarios.
- 3- Expand on the broad implications for bacterial pathogenesis and therapeutic methods.